# Regulation of eDHFR-tagged proteins with trimethoprim PROTACs

Jean M. Etersque [1,4], Iris K. Lee[1,4], Nitika Sharma[1,4], Kexiang Xu[1], Andrew Ruff[1], Justin D. Northrup [1], Swarbhanu Sarkar[1], Tommy Nguyen [1], Richard Lauman[2], George M. Burslem [2,3] & Mark A. Sellmyer [1,2] ✉

Temporal control of protein levels in cells and living animals can be used to improve our understanding of protein function. In addition, control of engineered proteins could be used in therapeutic applications. PRoteolysis-TArgeting Chimeras (PROTACs) have emerged as a small-molecule-driven strategy to achieve rapid, post-translational regulation of protein abundance via recruitment of an E3 ligase to the target protein of interest. Here, we develop several PROTAC molecules by covalently linking the antibiotic trimethoprim (TMP) to pomalidomide, a ligand for the E3 ligase, Cereblon. These molecules induce degradation of proteins of interest (POIs) genetically fused to a small protein domain, *E. coli* dihydrofolate reductase (eDHFR), the molecular target of TMP. We show that various eDHFR-tagged proteins can be robustly degraded to 95% of maximum expression with PROTAC molecule **7c**. Moreover, TMP-based PROTACs minimally affect the expression of immuno-modulatory imide drug (IMiD)-sensitive neosubstrates using proteomic and biochemical assays. Finally, we show multiplexed regulation with another known degron-PROTAC pair, as well as reversible protein regulation in a rodent model of metastatic cancer, demonstrating the formidable strength of this system. Altogether, TMP PROTACs are a robust approach for selective and reversible degradation of eDHFR-tagged proteins in vitro and in vivo.

Tools that allow conditional control of protein concentration are essential for understanding protein function. Several strategies have been developed to regulate proteins at different stages of the central dogma. For example, DNA-modifying technologies such as CRISPR/Cas9[1] and RNA interference-based techniques[2] (RNAi) modulate protein expression at the genomic and mRNA level, respectively. While both approaches are widely used given their relatively straightforward nature, they have limitations. Genome editing is generally irreversible and RNAi-based regulation depends on the inherent turnover rate of the protein of interest (POI), and therefore suffers from a relatively slow onset of action. This limits the temporal windows to modulate

stable proteins with a long half-life[3] and could enable proteome rewiring in response to the loss of POI, thus confounding biological discovery. Furthermore, these techniques present challenges for in vivo delivery and function, as they require viral vector or lipid nanoparticle-based distribution systems[4,5].

Protein degradation approaches that permit post-translational regulation of target proteins are desirable as they allow for fast, reversible, and dose-dependent reduction of POI abundance[6]. Different post-translational protein degradation strategies have been developed by leveraging the favorable pharmacokinetics of small molecules and can be grouped into two classes: i) those targeting

[1]Department of Radiology, Perelman School of Medicine, University of Pennsylvania, Philadelphia, PA, USA. [2]The Department of Biochemistry and Biophysics, Perelman School of Medicine, University of Pennsylvania, Philadelphia, PA, USA. [3]The Department of Cancer Biology, Perelman School of Medicine, University of Pennsylvania, Philadelphia, PA, USA. [4]These authors contributed equally: Jean M. Etersque, Iris K. Lee, Nitika Sharma. ✉e-mail: mark.sellmyer@pennmedicine.upenn.edu

endogenous proteins (requiring a unique high-affinity ligand) or ii) those targeting genetically engineered fusion proteins. One such tagging approach uses a protein destabilizing domain (DD), which targets POI-DD fusion proteins to the proteasome in the absence of a stabilizing ligand[7,8]. This represents a "drug-on" system as the fusion protein is expressed and functions normally only in the presence of a stabilizing ligand. The dose-dependent tunability and reversible feature of the DD system have been used successfully to study constitutively expressed proteins in many different organisms[9–12]. Another strategy is to use a "drug-off" system where the addition of ligand results in degradation of POI fused to a degron tag. Auxin-inducible degron (AID)[13] and ligand-inducible degradation (LID) domain[14,15] are two "drug-off" domains that use plant hormone auxin and Shield-1 ligand, respectively, to achieve degradation of tagged POIs. Bifunctional small molecules such as Proteolysis-Targeting Chimeras (PROTACs)[4], Autophagy-Targeting Chimeras (AUTACs)[5], and Lysosome-Targeting Chimeras (LYTACs)[16] are also examples of "drug-off" systems that have been developed to allow for targeted protein degradation of both endogenously and genetically-tagged proteins. Of these, PROTACs are the most extensively studied. ARV-110 and ARV-471 target endogenous proteins and have shown promise in clinical trials for prostate and breast cancer, respectively[17,18]. PROTACs contain one element that binds to the target protein, and another element that recruits an E3 ligase, to the target POI for degradation[18]. Given the difficulty of finding high-affinity ligands for the "undruggable" proteome, some high-affinity ligands have been repurposed to produce PROTACs targeting genetically engineered fusion proteins (as opposed to targeting endogenous proteins which require a unique ligand). These include HaloTag/PROTACs[19] and "dTAG" system based on a mutant cytosolic prolyl isomerase FKBP12$^{F36V}$ protein domain and its selective dTAG ligands[20,21]. Both have demonstrated robust proteasome-mediated degradation of their target proteins in vitro, although in vivo use has been limited.

*E. coli* dihydrofolate reductase (eDHFR) and its highly specific small molecule inhibitor, trimethoprim (TMP), is a protein tag-ligand pair that has been engineered for numerous biomedical applications. This is in part due to the well-known structure-activity relationship (SAR) and the nanomolar affinity of the ligand for the protein binding site[22–24]. For example, the Wandless group developed eDHFR as a DD for "drug-on" protein regulation[8]. Others have made bifunctional molecules containing TMP to achieve conditional protein dimerization for applications in protein localization and transcription regulation[25–27]. Furthermore, the Hedstrom group synthesized TMP-Boc$_3$Arg and used the hydrophobic *tert*-butyl groups on the molecule to tag and induce proteasome-mediated degradation of eDHFR, but not via a ubiquitin-dependent pathway[28]. However, limitations of hydrophobic tagging- particularly for in vivo use- include modest level of degradation observed (e.g., 20% of maximum expression), the need for a high concentration of ligand[28], and the poor pharmacological properties. Given the versatility of the eDHFR-TMP pair, including its application in positron emission tomography (PET) imaging to monitor cell-based therapies[29] and the development of fluorescent TMP derivatives for fluorescence-based assays[30], TMP-based PROTACs for regulation of eDHFR-fused proteins could be highly useful in diverse experimental settings.

Here, we report the development and evaluation of TMP PROTACs to regulate the expression of eDHFR-tagged proteins. Using linkers of different lengths, we covalently linked TMP to pomalidomide, a ligand for E3 ligase Cereblon, and identified a lead compound, **7c**. We tested **7c** for its capacity to degrade eDHFR fusions to various protein classes in several different cell lines including primary human cells, confirming its modular nature. We confirmed the mechanism of degradation, characterized the potential of **7c** to target proteins occupying different subcellular compartments, probed for off-target degradation using biochemical and proteomic assays, and

demonstrated the orthogonality to dTAG PROTACs. Finally, we evaluated **7c** in a rodent model of metastatic ovarian cancer, which showed promise for in vivo applications.

## Results

### In vitro regulation with TMP PROTACs

Genetic fusions can be expressed by adding the eDHFR coding sequence to the N- or C-terminus of a protein coding sequence, and the resultant fusion protein can be degraded with the addition of a TMP PROTAC (Fig. 1A). Given the precedents of synthesizing fluorescent and radiolabeled TMP derivatives[30–32], pomalidomide was added at the *para*-position of the trimethoxybenzene ring via flexible polyethylene glycol (PEG) linkers of variable length (Fig. 1B). Four active TMP PROTAC molecules (**7a-c** and **e**) were made and one control molecule with a methylated gluterimide nitrogen (**7f**) on the pomalidomide moiety (Supplementary Fig. 1). A PEG linker was chosen given its flexibility and membrane permeability, and is a standard motif used in the field in addition to alkyl chains[33–35]. Pomalidomide was chosen given its affinity for Cereblon, synthetic tractability, as well as its use in clinically-tested PROTACs[36].

To characterize the efficiency of degradation, we used two cell lines, JURKAT and HEK293T, which were transduced to express a fusion protein of eDHFR and yellow fluorescent protein (YFP), identified as JURKAT-eDHFR-YFP and HEK293T-eDHFR-YFP cells respectively (Fig. 2A). The construct also contained a C-terminal T2A-firefly Luciferase for orthogonal expression validation with bioluminescence. The ability of PROTACs (**7a-c** and **e**) to degrade eDHFR-YFP was evaluated in JURKAT-eDHFR-YFP cells for 4, 8, and 24 h time points. The data suggested that compounds **7b** and **7c** downregulate eDHFR-YFP most effectively, with degradation observed as early as 4 h post-treatment (Fig. 2B). To further validate this finding, the activity of the PROTACs was also assessed in HEK293T-eDHFR-YFP cells. Consistent with the results from JURKAT cells, **7a**, **7b** and **7c** showed the most effective degradation of eDHFR-YFP, in contrast to **7e** which did not show degradation of target, in the HEK293T cell line (Fig. 2C, D). Both in JURKAT-eDHFR-YFP and HEK294T-eDHFR-YFP cells, **7c** induced changes in eDHFR-YFP concentration in a dose-dependent fashion and demonstrated the expected "hook effect"[4,19,37,38](a U-shaped concentration dependence that is intrinsic to PROTAC activity), with optimal degradation was observed at a low nanomolar concentration of **7c**.

Since **7c** performed ideally in initial in vitro experiments, the drug metabolism/pharmacokinetic (DMPK) properties of **7c** were tested in mice (Supplementary Fig. 2). One of the DMPK analyses characterized blood plasma of male C57BL6 mice given **7c** by intravenous (IV), per os (PO) or intraperitonially (IP) delivery. The data showed that **7c** remains detectable in mouse blood plasma for up to 8 h when delivered via IP, suggesting feasibility of regulating protein with this molecule in an in vivo context. Thus, we persisted with further in vitro characterization of **7c** in cell models.

Next, the time and concentration dependence of **7c** to downregulate eDHFR-YFP in JURKAT cells was assessed. We observed 10% of maximum eDHFR-YFP fluorescence (90% degradation) in JURKAT-eDHFR-YFP cells with as low as 25 nM of **7c** at 12 h (Fig. 2E). Maximum downregulation of eDHFR-YFP in HEK293T-eDHFR-YFP cells was reached following 24 h of incubation with 25–100 nM of **7c** (Fig. 2F), noting, regulation appeared to be slightly delayed relative to the rapid downregulation observed in JURKATs. From these initial experiments, we demonstrated that the PROTAC **7c** can induce a dose-dependent downregulation of eDHFR-YFP in both JURKAT and HEK293T cells.

Subsequently, washout experiments were conducted to determine the reversibility of **7c**-induced downregulation of eDHFR-YFP. Both the JURKAT-eDHFR-YFP and HEK293T-eDHFR-YFP cells were incubated with 100 nM of **7c** for 24 h, washed with PBS three times to remove the compound, and were sampled at increasing time points

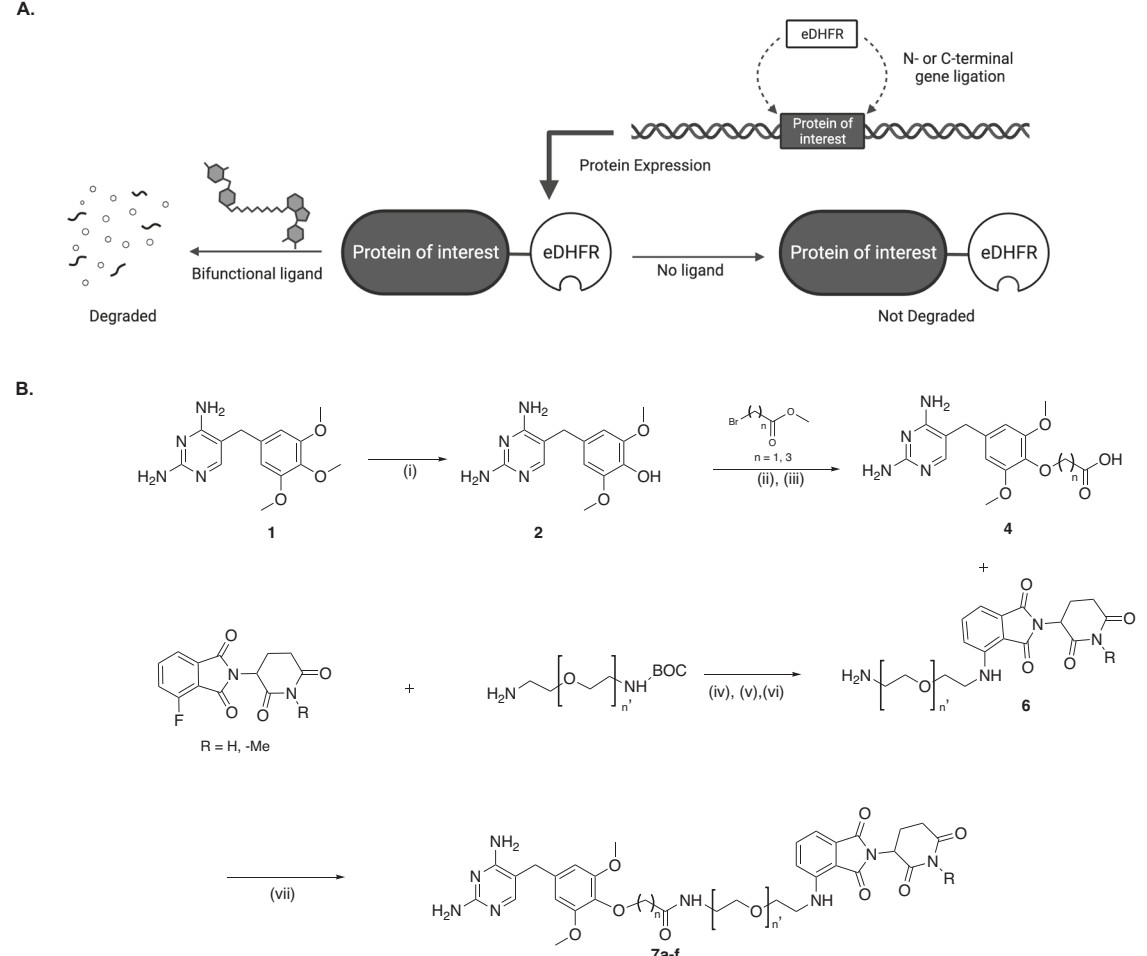

**Fig. 1 | Illustration of TMP-eDHFR PROTAC system and chemical approach.**
**A** The protein of interest (POI) is fused to the N- or C-terminus of eDHFR. The fusion protein is expressed in the absence of TMP PROTAC and is degraded in the presence of the regulating ligand via the proteasome. **B** Synthesis of TMP-pomalidomide PROTACs. (i) HBr, 90 °C, 20 min, 1 M NaOH; (ii) t-BuOK, DMSO, 2 h/ Cs$_2$CO$_3$, DMF, 70 °C, 12 h; (iii) K$_2$CO$_3$, MeOH, H$_2$O, 70 °C, 12 h; (iv) DIPEA, DMF, 90 °C, 12 h; (v) TFA, DCM, room temperature (rt), overnight; (vi) Cs$_2$CO$_3$, iodomethane, DMF; (vii) PyAOP, DIPEA, DMF, 30 min, rt.

following the washout to assess for recovery of eDHFR-YFP fluorescence as assessed by flow cytometry or Western blot analysis. Assessment by flow cytometry demonstrated that the eDHFR-YFP expression in JURKATs begins to return 3 h following the washout of **7c** and takes approximately 72 h for the signal to return to baseline (Fig. 2G). Similarly, in HEK293T cells, the eDHFR-YFP signal was 100% restored in 24 h following the removal of **7c** (Fig. 2H).

### Mechanism of eDHFR-YFP protein degradation with TMP PROTACs

To determine the mechanism of degradation induced by **7c**, HEK293T-eDHFR-YFP cells were pre-incubated with either proteasome inhibitor epoxomicin (500 nM) or lysosomal inhibitor hydroxychloroquine sulfate (HCS) (25 μM) for 1 h. Following pre-incubation, cells received either **7c** (100 nM), TMP (25 μM), or pomalidomide (2.5 μM) and were incubated for an additional 12 h. While cells treated with HCS and **7c** still exhibited robust degradation of eDHFR-YFP, we observed no reduction in eDHFR-YFP signal in cells treated with both proteasome inhibitor, epoxomicin, and **7c** (Supplementary Fig. 3A). A similar experiment was conducted using MLN4924 (500 nM) which blocks neddylation of Cul4 which is required for Cereblon's E3 ligase activity[4,39]. The effect of neddylation inhibition on **7c**-induced degradation of eDHFR-YFP was compared to that of autophagy inhibition with 3-methyladenine (3-MA) (25 μM). As anticipated, co-incubation of

cells with **7c** and MLN4924 prevented the degradation of eDHFR-YFP whereas blocking autophagy had no effect (Supplementary Fig. 3B). Together, these results confirm that **7c** facilitates the degradation of eDHFR-YFP via the ubiquitin-proteasome pathway, as do other PROTAC molecules that include pomalidomide.

In addition, we show that **7c**-induces the degradation of eDHFR-YFP by the formation of a ternary complex with the eDHFR-tagged target protein, **7c**, and E3 ligase protein Cereblon. HEK293T-eDHFR-YFP cells were treated with 25 μM TMP or 25 μM pomalidomide to block the binding sites on eDHFR and Cereblon respectively, directly followed by the addition of **7c** (100 nM). These data show, as expected, that **7c** alone will degrade eDHFR-YFP but in the presence of TMP or pomalidomide, it will not (Supplementary Fig. 3C). Next, an inactive analog of **7c** was synthesized in which the glutarimide nitrogen on pomalidomide was methylated (**7f**) as indicated by the synthetic scheme in Fig. 1B. This chemical modification is known to prevent interaction of the pomalidomide with the Cereblon binding pocket, preventing the ternary complex formation and subsequent degradation[40]. As expected, HEK293T-eDHFR-YFP cells treated with 100 nM of **7f** showed no eDHFR-YFP degradation (Supplementary Fig. 3D). Altogether, these experiments support that both eDHFR and Cereblon are required for the degradation of the target protein.

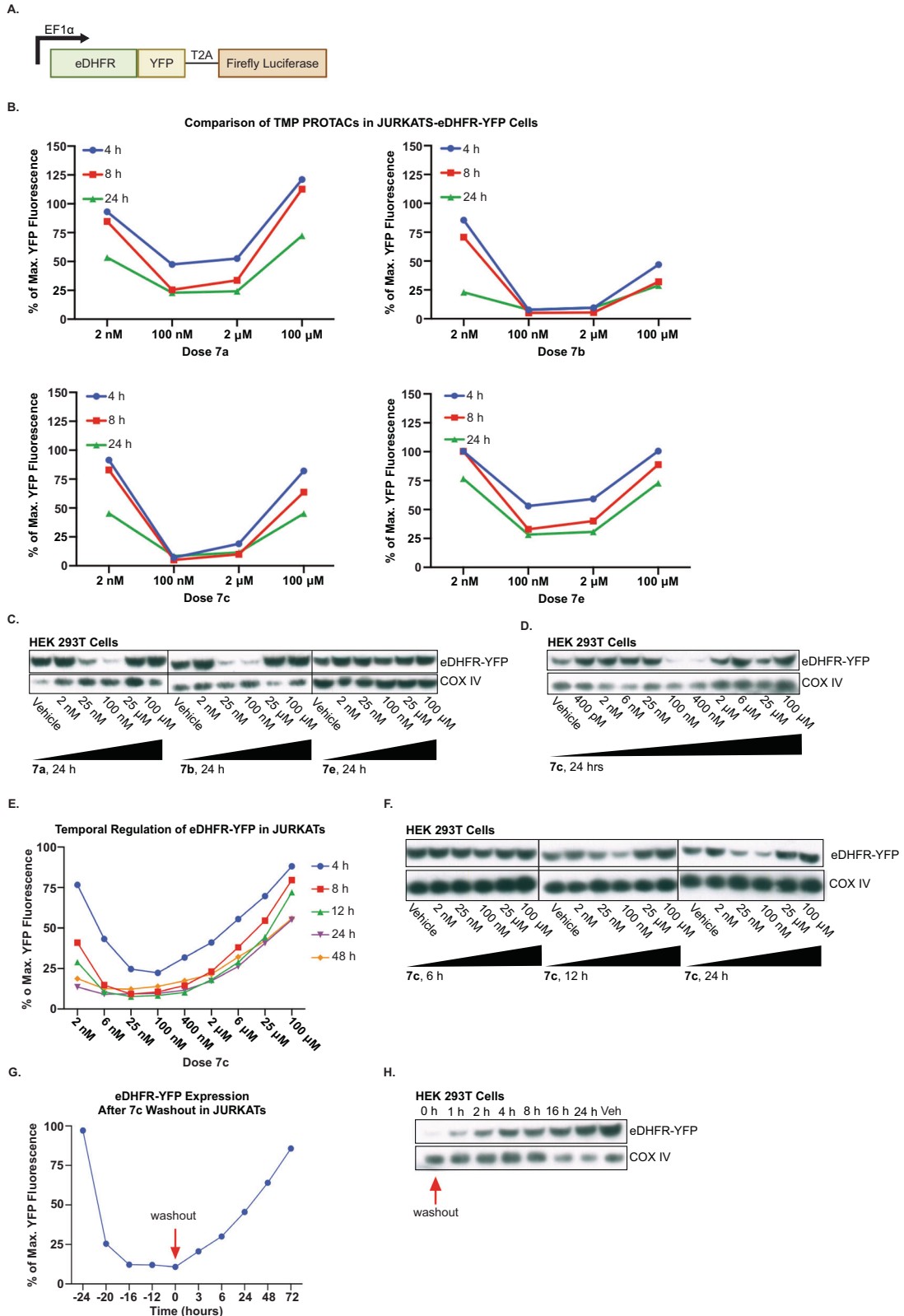

Even in cell lines in which expression of Cereblon E3 ligase is known to be low, degradation of target protein can still be modestly achieved. Incubation of HCT116 cells (Cereblon-low[41]) with **7c** resulted in ~40% decrease in luminescence signal of eDHFR-Luciferase after 24 h at 190 nM **7c**, but improved to ~70% decrease in luminescence at 20 nM **7c** after 48 h of incubation (Supplementary Fig. 4).

## TMP PROTAC specificity

Many studies have highlighted potential off-target effects of PROTACs that can lead to the degradation of unintended, non-target proteins[42–44]. This effect is particularly pronounced in PROTACs designed from immunomodulatory imide drugs (IMiDs), including pomalidomide, which have been shown to facilitate the interaction of

**Fig. 2 | Regulation of eDHFR-YFP in JURKAT and HEK293T cells with TMP PROTACs. A** Schematic of eDHFR-YFP-T2A-Luciferase construct. YFP is directly fused to the C-terminus of eDHFR to allow for regulation of the fluorescent protein with TMP PROTACs. **B** Screening of compounds **7a-c** and **e** in JURKAT-eDHFR-YFP cells. JURKAT-eDHFR-YFP cells were incubated with TMP-PROTACs for 4, 8, and 24 h. YFP fluorescence assessed by flow cytometry showed effective degradation of eDHFR-tagged YFP by **7b** and **7c**. Representative data from $n = 3$. **C** Dose-response screening of compounds **7a**, **7b**, and **7e** at 24 h post-incubation. Degradation of eDHFR-tagged YFP in HEK293T cells analyzed by Western blot with anti-GFP antibody. eDHFR-YFP protein fusion is 45 kDa. COX IV = loading control, 18 kDa. $n = 3$. **D** Dose–response of **7c** in HEK293T-eDHFR-YFP cells showed robust dose-dependent degradation of eDHFR-tagged YFP at 24 h post-incubation. eDHFR-YFP detected with anti-GFP antibody. eDHFR-YFP protein fusion is 45 kDa. COX IV = loading control, 18 kDa. $n = 3$. **E** Dose−response and time course of **7c** in JURKAT-eDHFR-YFP. Kinetics of YFP degradation by the lead compound **7c** was characterized by incubating JURKAT-eDHFR-YFP cells with various doses of **7c** for 4, 8, 12, 24, and 48 h and assessing for YFP expression with flow cytometry. Representative data from $n = 3$. **F** Dose−response and time course of **7c** in HEK293T-eDHFR-YFP cells at 6, 12, and 24 h post-incubation. eDHFR-YFP detected with anti-GFP antibody. eDHFR-YFP protein fusion is 45 kDa. COX IV = loading control, 18 kDa. $n = 3$. **G** Reversal kinetics of YFP degradation in JURKAT-eDHFR-YFP cells. JURKAT-eDHFR-YFP cells were incubated with 100 nM of **7c** and the YFP expression was monitored at several time points before and after drug washout. Representative data from $n = 3$. **H** Western blot analysis of YFP recovery in HEK293T-eDHFR-YFP cells incubated with 100 nM of **7c**. eDHFR-YFP detected with anti-GFP antibody. eDHFR-YFP protein fusion is 45 kDa. COX IV = loading control, 18 kDa. $n = 2$.

neo-substrates with Cereblon and induce non-specific degradation[45]. To characterize potential off-target effects of **7c**, we tested for any change in expression of Ikaros proteins IKZF1 and IKZF3 as well as a translation regulator, GSPT1, and Casein kinase 1, CK1α—all of which are well-known neo-substrates degraded by IMiDs[42]—following incubation with **7c**. We probed for these non-specific protein perturbations in two immortalized cell lines and primary human T cells.

In a proteomic experiment, HEK293T cells were treated with an ideal concentration for target regulation of **7c** (100 nM) for 48 h. No statistically significant changes in GSPT1 levels were seen (Supplementary Fig. 5A). Separately, we also conducted a dose–response in HEK293T cells with **7c** for 24 h and analyzed the cell lysates for expression of the neosubstrates by Western blot. These data indicated that CK1α expression in HEK293T-eDHFR-YFP cells was not affected at the optimal PROTAC doses for eDHFR-YFP degradation (25–100 nM) (Supplementary Fig. 5B). While CK1α and GSPT1 are known to be affected by lenalidomide[42,46], the pomalidomide-based TMP PROTAC, **7c**, did not target CK1α for degradation, however, by Western blot analysis, we observe some downregulation of GSPT1 expression at these concentrations (approximately 60% of max) (Supplementary Fig. 5B). In HEK293T-eDHFR-YFP cells, IKZF1 and IKZF3 levels were not detectable by Western blot. Lastly, we assessed whether **7c** treatment impacted cell viability and growth of HEK293T-eDHFR-YFP cells after 24 and 48 h of incubation with **7c**, which results show no changes in cell viability compared to vehicle control (Supplementary Fig. 5C).

In JURKAT cells and primary human T cells, IKZF1 and IKZF3 expression was investigated given the implication of the two proteins in T cell biology[47,48], as well as GSPT1 for its role in cell viability. To do this we again used mass spectrometry proteomics in primary human T cells stably expressing eDHFR-FLAG treated with **7c** for 24 h to assess for on- and off-target protein effects. As expected, there were statistically significant changes in eDHFR-FLAG, IKZF1 and IKZF3 protein levels, but GSPT1was not affected (Fig. 3A). As measured by Western blot in JURKAT-eDHFR-YFP cells, IKZF1 and IKZF3 were only modestly affected by **7c** at optimal TMP PROTAC doses (25–100 nM). At these concentrations of **7c**, IKZF1 showed approximately 100% of its maximum expression, whereas IKZF3 showed about 70% of maximum by densitometry (Fig. 3B, C); GSPT1 and CK1α could not be detected by Western blot. However, both IKZF1 and IKZF3 were clearly reduced at high doses (25 μM and 100 μM) of **7c**. Thus, **7c** has a feasible "therapeutic index" for selective degradation of target POI in the absence of confounding off-target degradation. To ensure that potential changes in protein concentration of non-targets influenced by **7c** do not affect cell viability of eDHFR-FLAG-positive primary human T cells, we conducted a dose–response-cell viability assay and determined that **7c** does not cause perturbations to cell viability (Fig. 3D).

### Regulation of various eDHFR protein fusions
Next, the generality of the eDHFR tag and **7c** pair was evaluated by regulating the expression of various classes of proteins. Proteins with differing subcellular locations were tested including a membrane-associated signaling molecule Lck, which is a lymphocyte-specific protein tyrosine kinase implicated in the formation of major histocompatibility complex (MHC) in immune cells, a secretory pathway-localized protein, interleukein-2 receptor subunit *beta*, known as CD122, and a nuclear protein runt-related transcription factor 1, RUNX1. The genes encoding these proteins were fused to the N-terminus of eDHFR by a Gly-Ser linker, followed by a C-terminal FLAG-tag for immunoblotting (Fig. 4A). HEK293T cells were transduced to express the fusion proteins and each cell line was incubated with **7c** to assess the level of target protein degradation. Immunoblotting demonstrated optimal degradation of Lck-eDHFR-FLAG, CD122-eDHFR-FLAG, and RUNX1-eDHFR-FLAG proteins with 25 nM-100 nM of **7c** following 24 h of incubation (Fig. 4B–D), underscoring the broad applicability of this molecular tool. Confocal fluorescence microscopy using a mouse Anti-FLAG antibody and Anti-mouse AlexaaFluor-488 and DAPI staining confirmed subcellular localization of the protein fusions (Fig. 4E, Supplementary Fig. 6).

### Multiplexed regulation with orthogonal PROTAC systems
Next, we evaluated the possibility of our PROTAC-tag system being paired with other PROTAC-tag systems to achieve multiplexed protein regulation in a single cell. There are biological applications of degrading multiple tagged proteins simultaneously i.e., to probe synthetic lethality or the protein redundancy. Since other approaches for tag-based degradation have been explored previously (e.g. FKBP12$^{F36V}$/dTAGv-1[20,21] and HaloTag/HaloPROTAC[19]), we sought to investigate whether eDHFR-TMP could operate in concert with existing technologies. A genetic construct of mCherry fused to FKBP12$^{F36V}$ (mCherry-FKBP$^{F36V}$) and EGFP fused to eDHFR (EGFP-eDHFR) with a P2A cleavage site in-between (Fig. 5A) was generated and stably expressed in OVCAR8 cells. Cells were then treated with various doses of **7c**, dTAGv-1, or both PROTAC ligands for 24 h and analyzed by Western blot. The data show that treatment with **7c** downregulates EGFP-eDHFR between 25 nM-100 nM while not affecting mCherry-FKBP12$^{F36V}$ abundance (Fig. 5C). Treatment with dTAGv-1 knocked down mCherry-FKBP12$^{F36V}$ abundance at concentrations 25 nM and higher, without influencing EGFP-eDHFR expression (Fig. 5D). Finally, treatment with both drugs downregulated both constructs to nearly 100% depletion compared to vehicle controls (Fig. 5E). Altogether, these data demonstrate that multiple PROTAC-tag systems can be used simultaneously to affect disparate protein abundances in the same cell population.

### In vivo degradation of eDHFR-Luc in a mouse model
To test the feasibility of the eDHFR-TMP PROTAC system to regulate proteins in vivo, OVCAR8 were transduced to express eDHFR-firefly luciferase fusion protein (eDHFR-Luc), bioluminescence applications in animals, and a T2A-mCherry gene for expression validation (Fig. 5A). OVCAR8-eDHFR-Luc cells were treated with **7c** at various doses and time points to understand the kinetics and efficiency of degradation

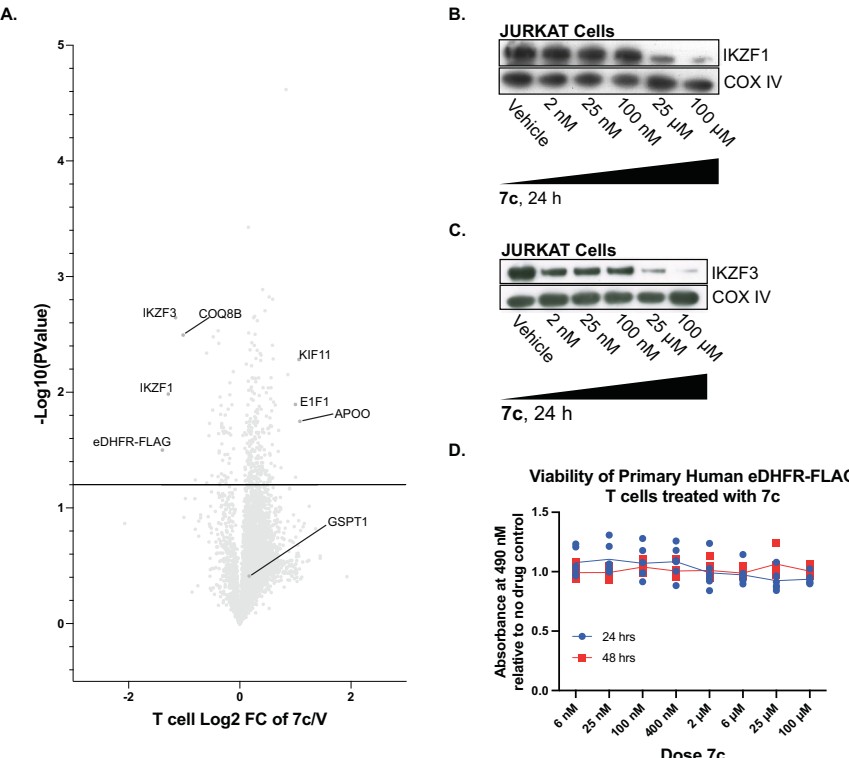

**Fig. 3 | Characterization of IMiD-sensitive proteins and cell viability in primary human T cells and JURKATs. A** Mass spectrometry proteomics of cell lysates from primary human T-eDHFR-FLAG cells treated with 100 nM **7c** for 24 h. Volcano plot shows effect of **7c** on protein levels in primary human T-eDHFR-FLAG cells relative to vehicle control. $n = 2$, each experiment with 4 technical replicates, data point represents mean value. Statistical significance was determined using a two-sided Student's $t$ test. **B** Dose–response characterization of off-target binding and degradation of IKZF1 in JURKAT-eDHFR-YFP cells. IKZF1 detected with anti-Ikaros antibody, 53 kDa. COX IV = loading control, 18 kDa. $n = 2$. **C** Dose–response characterization of off-target binding and degradation of IKZF3 in JURKAT-eDHFR-YFP cells. IKZF3 detected with anti-Aiolos antibody, 58 kDa. COX IV = loading control, 18 kDa. $n = 3$. **D** Dose–response-cell viability characterization of primary human T cells expressing eDHFR-FLAG when treated with **7c** for either 24 or 48 h compared to no drug control. $n = 1$, data points are mean ± SD, 6 technical replicates.

(Fig. 5B). After 12 h, **7c** induced peak degradation of the eDHFR-Luc (90% signal reduction) in OVCAR8 cells, with evidence of target degradation in as early as 4 h. Given the favorable findings shown in OVCAR8-eDHFR-Luc cells this model system was used to assess **7c** function in vivo. OVCARs are a human ovarian tumor line that can be engrafted along the mouse peritoneum, mimicking human metastatic ovarian carcinoma. In brief, $10 \times 10^6$ eDHFR-Luciferase OVCAR8 cells were injected intraperitoneally into nude mice and the tumors were grown for 4 weeks before administering **7c**. Pilot experiments and the DMPK profile of **7c** suggested that repeated dosing would be necessary to sustain the concentration of **7c** in the plasma (Supplementary Fig. 2A–D)[20,21]. Therefore, a model of TID dosing (3x per day) was performed for one day and bioluminescent imaging (BLI) was used to monitor tumor eDHFR-Luc expression both before and after treatment with **7c** (Fig. 5C). Quantification of the BLI images showed a statistically significant ~4-fold drop in eDHFR-Luc signal 2-3 days after **7c** administration (Fig. 5D). This depleted signal on day 3 (Fig. 5E), returned to near baseline by day 7 and 9 (Fig. 5C, D). In addition, a control using an admixture of TMP and pomalidomide (not covalently linked) in the vehicle was performed, which did not show a statistically significant decrease in luminescence compared to **7c** (Supplementary Fig. 7).

## Discussion

Ligand-mediated degradation of a protein target is an effective molecular strategy for dose-dependent, temporal control of endogenous protein activity. While there are benefits to this method of protein regulation, the creation of small molecule ligands for each new target is time-consuming and often not achievable if the protein does not possess a ligand binding site that affords high specificity. Applying degron protein tags to POIs is an effective method for targeted molecular regulation and circumvents the need to develop novel ligands for the protein target of interest. With advances in cell engineering[49] and mRNA-mediated in vivo engineering[50], using protein fusions may also prove to be a viable strategy for regulating therapeutic targets.

Here, we developed TMP-based PROTACs to control the cellular abundance of eDHFR-tagged fusion proteins. eDHFR is an ideal protein tag for the regulation of partner fusion proteins because it is small (18 kDa) and already has applications in both, in vitro and in vivo, imaging. The Cornish group has made fluorescent derivatives of TMP[30], which can be used in fluorescence microscopy experiments, and our group has developed TMP-based radiotracers for positron emission tomography (PET), which provide sensitive and quantitative measures of engineered cells within tissues in animal models[29,32]. We envision that investigators will be able to make measurements on the locations of eDHFR tagged proteins using such imaging techniques and then use TMP PROTACs for target regulation at the appropriate time.

To develop a TMP and pomalidomide-based PROTAC, we varied the linker length substantially to achieve optimal degradation. Compounds **7b** and **7c** showed better degradation kinetics and profile than **7a** and **7e** (Fig. 2B–D), and **7c** was chosen for both its superior efficacy and ideal pharmacological properties. TMP-based ligands specifically bind to eDHFR with little to no interaction with the endogenous DHFR in mammalian cells due to the difference in binding affinity for the homologous proteins, where the $K_d$ of TMP for mammalian DHFR is

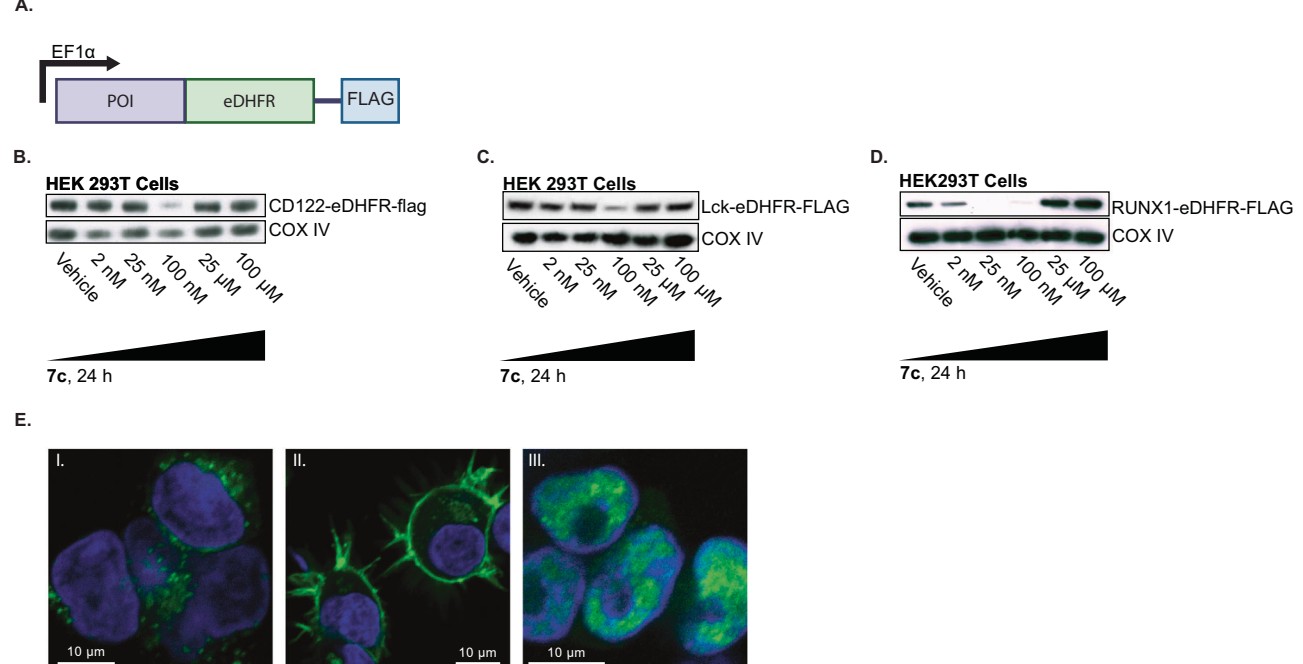

**Fig. 4 | Regulation of various eDHFR-fusion proteins with 7c. A** Schematic of POI-eDHFR fusion construct. POI was directly fused to the N-terminus of eDHFR to allow for regulation with **7c**. FLAG-tag was cloned downstream of eDHFR (to its C-terminus) for easy detection with anti-FLAG antibody. POI represents Lck, CD122, and RUNX1. **B** HEK293T-POI-eDHFR cells were incubated with **7c** for 24 h. Western blot analysis was performed to probe for the expression of CD122, $n = 3$, **C** Lck, $n = 3$, and **D** RUNX1, $n = 3$, all detected with anti-FLAG antibody. CD122-eDHFR-FLAG fusion protein is 81 kDa, Lck-eDHFR-FLAG is 77 kDa, and RUNX1-eDHFR-FLAG is 73 kDa. COX IV = loading control, 18 kDa. **E** Immunofluorescence of cells expressing POI-eDHFR-FLAG constructs with AlexaFluor-488 secondary antibody against Anti-FLAG primary antibody (green), and DAPI nuclear staining (blue). From left to right: I. CD122-eDHFR-FLAG localized in the secretory pathway, II. LCK-eDHFR-FLAG membrane associated, and III. RUNX1-eDHFR-FLAG is in the nucleus. $n = 3$.

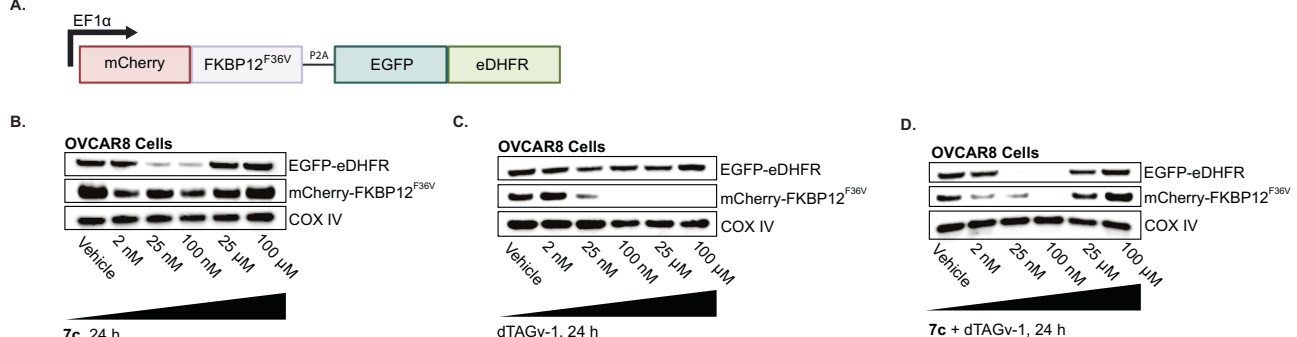

**Fig. 5 | Dual regulation of proteins with TMP and dTAG PROTACs in OVCAR8 cells. A** Schematic of genetic construct containing mCherry-FKBP12$^{F36V}$ and EGFP-EDHFR, with a cleavable P2A site in-between to ensure proportionally expression of each fusion protein. **B** OVCAR8-mCherry-FKBP12$^{F36V}$-EGFP-EDHFR cells were incubated with **7c** for 24 h and cell lysates were isolated for Western Blot analysis (see methods). mCherry-FKBP12$^{F36V}$ detected with anti-mCherry antibody and EGFP-eDHFR detected with anti-GFP antibody on the same membrane. mCherry-FKBP12$^{F36V}$ protein fusion is 40 kDa and eDHFR-YFP protein fusion is 45 kDa. COX IV = loading control, 18 kDa. $n = 3$ **C** OVCAR8-mCherry-FKBP12$^{F36V}$- EGFP-EDHFR cells were incubated with dTAGv-1 for 24 h. mCherry-FKBP12$^{F36V}$ detected with anti-mCherry antibody and EGFP-eDHFR detected with anti-GFP antibody on the same membrane. mCherry-FKBP12$^{F36V}$ protein fusion is 40 kDa and eDHFR-YFP protein fusion is 45 kDa. COX IV = loading control, 18 kDa. $n = 3$ **D** OVCAR8-mCherry-FKBP12$^{F36V}$-EGFP-EDHFR cells were incubated with **7c** and dTAG-1 for 24 h. mCherry-FKBP12$^{F36V}$ detected with anti-mCherry antibody and EGFP-eDHFR detected with anti-GFP antibody on the same membrane. mCherry-FKBP12$^{F36V}$ protein fusion is 40 kDa and eDHFR-YFP protein fusion is 45 kDa. COX IV = loading control, 18 kDa. $n = 3$.

approximately 4 orders of magnitude greater than that of eDHFR[22–24]. This difference, and that TMP's affinity for eDHFR is approximately 100x greater than pomalidomide's affinity for CRBN, enhances the "therapeutic window," of **7c** and its specificity towards eDHFR-tagged proteins. Specificity is important in IMiD sensitive cells lines, such as JURKATs or primary immune cells, where protein neosubstrates are not substantially affected by **7c**, but complete degradation of eDHFR-

tagged proteins is still achieved with nanomolar concentrations of drug (Fig. 3B). Proteomic experiments showed regulation of proteins as expected, such that eDHFR-YFP in HEK293T cells and eDHFR-FLAG in primary human T cells were downregulated by **7c** relative to vehicle control with statistical significance (Fig. 3A and Supplementary Fig. 5A). Furthermore, we found that GSPT1 abundance was not affected in either cell line (Fig. 3A and Supplementary Fig. 5A),

however, there was downregulation of IKZF1 and IKZF3 in T cells (Fig. 3A–C). Finally, we validated that treatment of HEK293T-eDHFR-YFP cells and eDHFR-FLAG+ primary human T cells with **7c** does not influence cell viability (Fig. 3D and Supplementary Fig. 5C).

**7c** showed effective degradation of various POIs—including YFP and Luciferase, as well as Lck, CD122, and RUNX1—in different cell types, with doses between 25 nM-100 nM (Fig. 2B–D, Fig. 4B–D and Fig. 6B and Supplementary Fig. 6). As anticipated, TMP PROTACs demonstrated a "hook effect" dose–response where the ternary complex of proteins and ligand is disfavored at high concentrations of PROTACs[4,19,37,38].

The activity of our lead molecule **7c** is comparable to that of previously developed PROTAC-tag systems[19,20,51]. For example, Halo-Tag is derived from a bacterial dehalogenase[19], and was adapted as a PROTAC-degron system using a peptide ligand functionalized with a von Hippel−Lindau (VHL) E3 ligase-targeting moiety and a chloroalkane tail (HaloPROTACs)[19]. While HaloPROTACs demonstrated a greater than 90% reduction in the abundance of various HaloTag-tagged POIs, the most effective, near complete degradation was observed at 500 nM of HaloPROTAC 3[21,38], which is higher than the effective dose range of **7c** (25–100 nM). When assessing the FKBP12^{F36V}-dTAG system, variability in dose–response between different fusion proteins using first-generation dTAG molecules was observed[20]. Although second-generation dTAG ligands that target VHL E3 ligase show specific target activity and high-fidelity degradation, solubility and formulation of the ligand remain a challenge[20,21]. Both HaloPROTACs and FKBP12^{F36V}-dTAG have demonstrated robust proteasome-mediated degradation of most target proteins in vitro, although in vivo use has not been easily or rapidly adopted[19–21,38]. Furthermore, the ligands used for both HaloTag and FKBP12^{F36V} systems are limited for imaging applications compared to TMP[51–53]. For example, while the metalated HaloPET radioligand has shown some uptake in mice, the radiotracer exhibited poor in vivo pharmacokinetic (PK) and pharmacodynamic (PD) profiles due to its hydrophobic nature, leading to low target-to-background contrast[52]. Similarly, for the FKBP12^{F36V} system, Shield-1-based radiotracers were considered, but it may also be difficult to achieve high contrast imaging based on a single amino acid change (F36V) from endogenous FKBP[53].

Beyond the attractive potential applications of our tool for companion imaging and regulation, this work is an important demonstration of the use of two PROTAC-tag systems, one targeting Cereblon, the other, VHL, in the same cell line to achieve dual regulation of two independent proteins. This approach can be potentially applied to probe for synthetic lethality, the coordination of proteins in protein complexes and signaling cascades within a cell. We showed orthogonal regulation of proteins in OVCAR8 cells express an eDHFR-fused protein (EGFP-eDHFR) and a FKBP-fused protein (mCherry-FKBP^{F36V}) that are targeted by the corresponding ligands, **7c** and dTAGv-1. Treatment of cells with **7c** showed robust knockdown of EGFP-eDHFR at the expected concentrations of 25 nM-100nM (Fig. 5B), where treatment of cells with dTAGv-1 selectively knocked down mCherry-FKBP^{F36V}, with no change in the level of EGFP-eDHFR expression (Fig. 5C).

We show strong, reversible, BLI signal loss related to **7c** administration in vivo. While FKBP^{F36V}-dTag has shown differences in luciferase signal compared to vehicle control, it did not produce a functional decrease in signal[20], where as **7c** decreased luciferase signal by 4-fold. In addition, TMP PROTACs could rapidly scaled for animal experiments given inexpensive starting materials and favorable solubility in PEG/water formulations. We envision potential expansion of this approach to control therapeutic proteins and thereby regulate therapeutic outcomes, perhaps even clinically bound therapeutics. For example, there is a need to regulate genetic and cellular medicines, such as chimeric antigen receptor (CAR) T cells, given their inherent self-regulating behaviors as "living drugs" and potential for on-target/off-tumor trafficking that could lead to toxicities[54–56]. The ability to control CAR expression and thereby titrate the degree of CAR T cell activity would be of significant benefit as it could not only help mitigate toxicities but also allow for individualized tuning of CAR T cell activity depending on the patient's degree of immune response.

A limitation of the study is that the functional outcomes of achieved regulation was not tested for several of the fused proteins (Lck-, CD122-, RUNX1-), beyond noting that the localization of these different fusion proteins was retained. eDHFR as a tag for endogenous proteins, has some attractive characteristics (e.g., small size, regulation, and imaging capabilities) and is a future area of investigation. Furthermore, while we have optimized for in vitro efficacy, the in vivo PK of TMP PROTACs may need to be further optimized, especially in terms of creating new linkers with improved pharmacological properties.

In summary, TMP PROTACs paired with the eDHFR tag, is a compelling approach to conditionally control protein expression, especially as it has clear potential for orthogonal protein regulation with existing technologies, as well as multi-modality imaging and in vivo advantages.

## Methods

All research completed complied with relevant ethical regulations and institutional approval. All recombinant DNA work was performed under Institutional Biosafety Committee (IBC) approval. The University of Pennsylvania-University Laboratory Animal Resources (ULAR) and Institutional Animal Care and Use Committee (IACUC) organizations approved the use of vertebrate animals for this study. We followed the ULAR/IACUC approved protocol (Sellmyer 805477-aaeifcb) and employed best laboratory practices to ensure the safety of the animals and researchers involved in this study. These protocols align with the ethics standards set forth by the Animal Welfare Act (AWA) and the "Guide for the Care and Use of Laboratory Animals."

### Chemical procedures and materials

Unless otherwise noted, chemicals were purchased from commercial suppliers at the highest purity grade available and were used without further purification. Thin layer chromatography was performed on 0.25 mm silica gel plates (60F254) using UV light as the visualizing agent. Silica gel (100−200 mesh) was used for column chromatography. Nuclear magnetic resonance spectra were recorded on a 400 MHz spectrometer, and chemical shifts are reported in δ units, parts per million (ppm). Spectra were referenced internally to the residual proton resonance in CDCl$_3$ (δ 7.26 ppm), Methanol-d4 (δ 4.78 ppm), or with tetramethylsilane (TMS, δ 0.00 ppm) as the internal standard. Chemical shifts (δ) were reported as part per million (ppm) on the δ scale downfield from TMS. $^{13}$C NMR spectra were referenced to CDCl$_3$ (δ 77.0 ppm, the middle peak) and Methanol-d4 (δ 49.3 ppm). Coupling constants were expressed in Hz. The following abbreviations were used to explain the multiplicities: s = singlet, d = doublet, t = triplet, m = multiplet. High resolution mass spectra were recorded with a micro TOF-Q analyzer spectrometer by using the electrospray mode. Target compounds and/or intermediates were characterized by liquid chromatography/mass spectrometry (LCMS) using a Waters Acquity separation module. Abbreviations used: DCM for dichloromethane, DMF for *N,N*-dimethylformamide, DMSO for dimethyl sulfoxide, DIPEA for *N,N*-diisopropylethylamine, MeOH for methanol, NaOH for sodium hydroxide, *t*-BuOK for potassium tert-butoxide, HBr for hydrobromic acid.

*Synthesis of 4-((2,4-diaminopyrimidin-5-yl)methyl)−2,6-dimethoxyphenol (2):* trimethoprim (5.00 g, 17.12 mmol) was dissolved in HBr (62 mL, 48% in H$_2$O), stirred at 95 °C for 30 min, and then quenched by slow addition of 12 mL 50% NaOH. The reaction mixture was allowed to cool to room temperature (RT), and placed at 4 °C overnight, allowing crystals to form. The precipitate was filtered and washed with ice-cold water. The collected precipitate was dissolved in boiling H$_2$O and 1 N NaOH was added to neutralize, leading to recrystallization. Crystals were washed with water and filtered under vacuum to afford 4-((2,4-

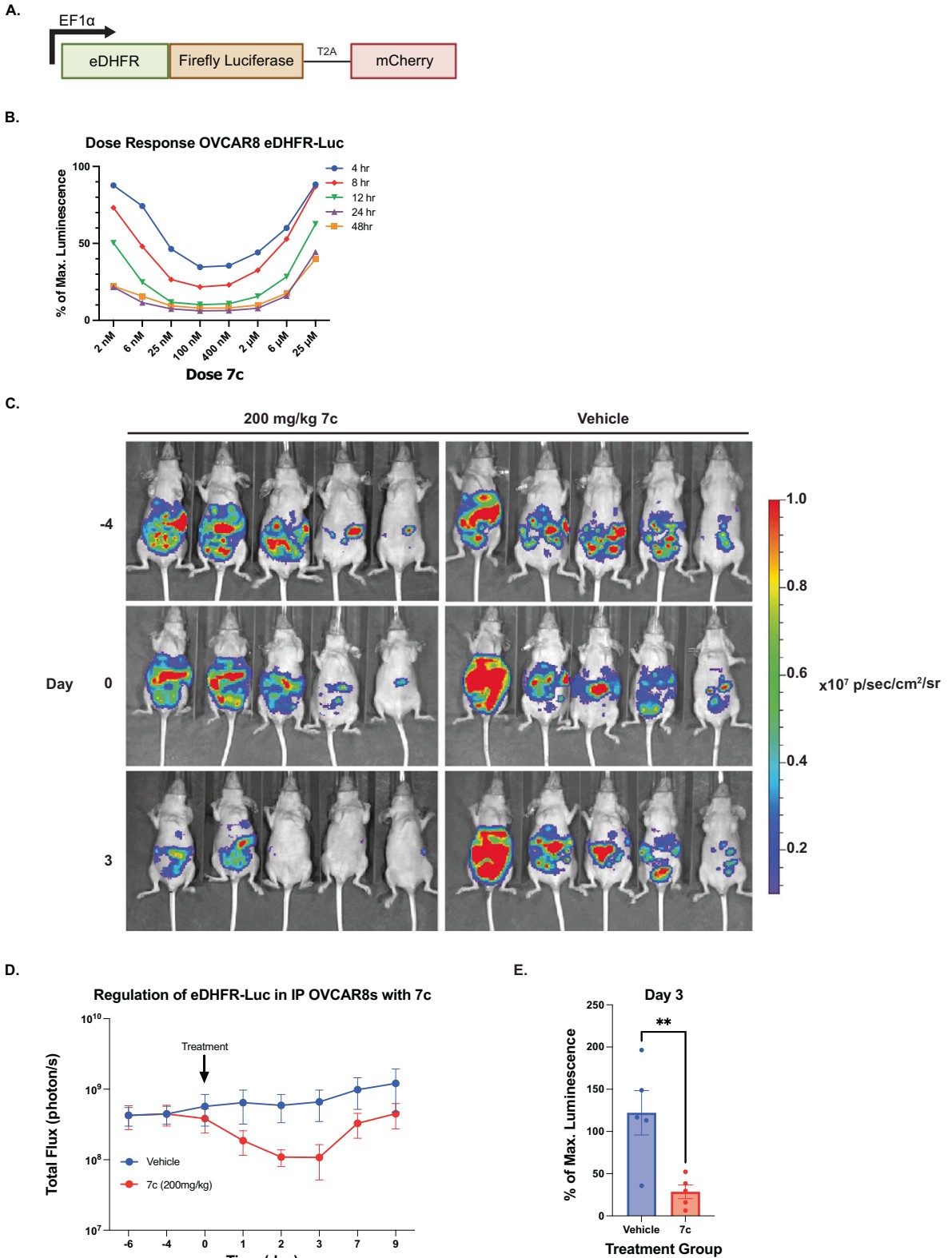

**Fig. 6 | Regulation of eDHFR-Luciferase expression in vivo using IP OVCAR8 cell tumor model. A** Schematic of eDHFR-Luciferase direct fusion construct. Luciferase was directly fused to the C-terminus of eDHFR to allow for regulation of the luminescent protein with **7c. B** OVCAR8-eDHFR-Luciferase cells were incubated with **7c** for 4, 8, 12, 24, and 48 h. Representative data from $n = 3$. **C** $10 \times 10^6$ OVCAR8-eDHFR-Luc cells were injected intraperitonely (IP) in CD-1 nu/nu mice. Following 4 weeks of tumor growth, BLI was performed to measure baseline Luciferase expression (Day −4). On Day 0, 200 mg/kg of **7c** (or vehicle control) was administered IP 3 times every 3 h, and BLI was performed 1 h after the last dose and monitored for 9 days. $n = 5$ for each group, data points are mean ± SEM. Animals from the D-4, D0, and D3 are shown. **D** Quantification over time of average BLI signal for mice given 200 mg/kg of **7c** compared to vehicle control as described in (**C**). **E** Percent (%) of maximum luminescence from BLI on Day 3 post-treatment. Data was normalized to baseline luminescence signal on Day 0 and groups were compared using a two-tailed unpaired $t$ test. **$p$-value = 0.0093. $n = 5$ for each group, data points are mean ± SEM.

diaminopyrimidin-5-yl)methyl)−2,6-dimethoxyphenol **2** (3.2 g, 68%) as a white solid[31]. LCMS(ESI); m/z: [M + H]$^+$calcd. for C$_{13}$H$_{17}$N$_4$O$_3$, 277.13; Found 277.35.

*Synthesis of methyl 2-(4-((2,4-diaminopyrimidin-5-yl)methyl)−2,6-dimethoxyphenoxy)acetate (**3a**):* t-BuOK (530 mg, 4.7 mmol, 1.1 equiv.) was added to a solution of **2** (1.2 g, 4.3 mmol) in anhydrous DMSO (25 mL) while stirring under Ar atmosphere. The solution was stirred at RT for a few minutes and turned to deep orange. Methyl bromoacetate (0.452 mL, 4.7 mmol, 1.1 equiv.) was added to the solution and the reaction was stirred at RT for 2 h. The reaction was monitored by TLC (10% MeOH/DCM) and following completion of the reaction, the solvent was removed under reduced pressure and the residual brown oil was subjected to column chromatography on silica gel with elution with 5-10% CH$_2$Cl$_2$/CH$_3$OH to afford **3a** (700 mg, 47%) as a whitish-brown solid[57]. LCMS(ESI); m/z: [M + H]$^+$calcd. for C$_{16}$H$_{21}$N$_4$O$_5$, 349.15; Found 349.28.

*Synthesis of methyl 4-(4-((2,4-diaminopyrimidin-5-yl)methyl)−2,6-dimethoxyphenoxy)butanoate (**3b**):* Cesium carbonate (2.8 g, 8.6 mmol, 2 equiv.) was added to the solution of **2** (1.2 g, 4.3 mmol) in anhydrous DMF (40 mL). The mixture was allowed to stir at RT for a few minutes and the color changed to deep orange. Methyl 4-bromo butanoate (0.778 g, 4.3 mmol) was added to the solution and the reaction was stirred at 70 °C overnight. The reaction completion was monitored by TLC (10% Methanol/DCM) and DMF was removed under a high vacuum following reaction completion. Water was added to the residue and the solution was extracted with ethyl acetate (2×50 mL). The organic layer was washed with aqueous sodium bicarbonate solution, dried over sodium sulfate, and the solvent was removed under reduced pressure. Trituration with isopropyl ether afforded **3b** (1.1 g, 69%) as a light brown solid[58]. LCMS(ESI); m/z: [M + H]$^+$calcd. for C$_{18}$H$_{25}$N$_4$O$_5$, 377.18; Found 377.44.

*Synthesis of 2-(4-((2,4-diaminopyrimidin-5-yl)methyl)−2,6-dimethoxyphenoxy)acetic acid (**4a**):* Potassium carbonate (869 mg, 6.3 mmol, 4.5 equiv.) was added to the solution of methyl 2-(4-((2,4-diaminopyrimidin-5-yl)methyl)−2,6-dimethoxyphenoxy)acetate **3a** (500 mg, 1.4 mmol) in methanol (10 mL), followed by addition of water (4 mL). The reaction was heated at 60 °C overnight. Methanol was evaporated under reduced pressure, and water was added (25 mL) to the residue. Neutrals were removed by extraction with ethyl acetate (2 × 50 mL). The aqueous layer was neutralized with 6 M HCl to pH ~7. It was concentrated to ~10–15 mL and left at 4 °C overnight. Filtration the next day afforded **4a** (332 mg, 71%) as light brown solid which was pure to proceed to next step. LCMS(ESI); m/z: [M + H]$^+$ calcd. for C$_{15}$H$_{19}$N$_4$O$_5$, 335.14; Found 335.34.

*Synthesis of 4-(4-((2,4-diaminopyrimidin-5-yl)methyl)−2,6-dimethoxyphenoxy)butanoic acid (**4b**):* Potassium carbonate (1.6 g, 11.7 mmol, 4.5 equiv.) was added to the solution of methyl 4-(4-((2,4-diaminopyrimidin-5-yl)methyl)−2,6-dimethoxyphenoxy)butanoate **3b** (1 g, 2.6 mmol) in methanol (18 mL), followed by addition of water (6 mL). The reaction was heated at 60 °C overnight and the steps outlined under the synthesis of **4a** were repeated. Filtration afforded **4b** (706 mg, 75%) as a light brown solid which was pure to proceed to next step. LCMS(ESI); m/z: [M + H]$^+$calcd. for C$_{17}$H$_{23}$N$_4$O$_5$, 363.17 Found 363.34.

*Synthesis of tert-butyl (2-(2-((2-(2,6-dioxopiperidin-3-yl)−1,3-dioxoisoindolin-4-yl)amino)ethoxy)ethyl)carbamate (**5a**):* Using a literature method[59], N-Boc-2-(2-Aminoethoxy)ethanamine (306 mg, 1.5 mmol) was added to the solution of 2-(2,6-dioxopiperidin-3-yl)−4-fluoroisoindoline-1,3-dione (276 mg, 1 mmol) in DMF (3 ml), followed by DIPEA (0.7 mL, 4 mmol, 4 equiv.). The reaction mixture was heated at 90 °C for 12 h. After reaction completion (monitored by the TLC), the dark green reaction mixture was poured into water (10 mL) and extracted with ethyl acetate (2 × 10 mL). The combined organic layer was washed with brine, dried over sodium sulfate, and the solvent was removed under reduced pressure. The residue was purified by column chromatography to afford **5a** in (262 mg, 57%) as yellow-green syrup.

*Synthesis of 2-(tert-butoxy)-N-(2-(2-(2-((2-(2,6-dioxopiperidin-3-yl)−1,3-dioxoisoindolin-4-yl)amino)ethoxy)ethoxy)ethyl)acetamide (**5b**):* The procedure was analogous to that described for compound **5a**. 2-(2,6-dioxopiperidin-3-yl)−4-fluoroisoindoline-1,3-dione (276 mg, 1 mmol) and tert-butyl (2-(2-(2-aminoethoxy)ethoxy)ethyl)carbamate (372 mg, 1.5 mmol) as starting materials afforded **5b** (300 mg, 60%) as yellow-green syrup.

*Synthesis of 4-((2-(2-aminoethoxy)ethyl)amino)−2-(2,6-dioxopiperidin-3-yl)isoindoline-1,3-dione (**6a**):* TFA(1.1 mL, 1.42 mmol, 2.5 equiv.) was added to the solution of 2-(tert-butoxy)-N-(2-(2-(2-((2-(2,6-dioxopiperidin-3-yl)−1,3-dioxoisoindolin-4-yl)amino)ethoxy)ethoxy)ethyl)acetamide **5a** (262 mg, 0.57 mmol) in DCM (7 mL), and the reaction mixture was stirred at RT for 5 h. After reaction completion (monitored by TLC), DCM (20 mL) was added to the mixture. The organic layer was washed with sodium carbonate solution, dried over sodium sulfate, and the solvent was removed under reduced pressure to obtain **6a** (174 mg, 85%) as yellow solid. LCMS(ESI); m/z: [M + H]$^+$calcd. for C$_{17}$H$_{21}$N$_4$O$_5$, 361.15; Found 361.32.

*Synthesis of 4-((2-(2-(2-aminoethoxy)ethoxy)ethyl)amino)−2-(2,6-dioxopiperidin-3-yl)isoindoline-1,3-dione (**6b**):* The procedure was analogous to that described for compound **6a**. 2-(tert-butoxy)-N-(2-(2-(2-((2-(2,6-dioxopiperidin-3-yl)−1,3-dioxoisoindolin-4-yl)amino)ethoxy)ethoxy) ethyl)acetamide **5b** (287 mg, 0.57 mmol) as starting material afforded **6b** (200 mg, 87% yield) as yellow solid. LCMS(ESI); m/z: [M + H]$^+$calcd. for C$_{19}$H$_{25}$N$_4$O$_6$, 405.18; Found 405.44.

*Synthesis of 2-(4-((2,4-diaminopyrimidin-5-yl)methyl)−2,6-dimethoxyphenoxy)-N-(2-(2-(2-((2-(2,6-dioxopiperidin-3-yl)−1,3-dioxoisoindolin-4-yl)amino)ethoxy)ethoxy)ethyl)acetamide (**7a**):* To the solution of 2-(4-((2,4-diaminopyrimidin-5-yl)methyl)−2,6-dimethoxyphenoxy)acetic acid **4a** (50 mg, 0.15 mmol) in DMF (2 mL) was added 4-((2-(2-(2-aminoethoxy)ethoxy)ethyl)amino)−2-(2,6-dioxopiperidin-3-yl)isoindoline-1,3-dione **6b** (61 mg, 0.15 mmol) followed by DIPEA (14.6 mg, 0.11 mmol, 0.75 equiv.) and PyAOP (98 mg, 0.19 mmol, 1.27 equiv.). The reaction mixture was stirred at RT for half an hour. After reaction completion as monitored by the TLC, the reaction mixture was poured into water and extracted with DCM (2×10 mL). The combined organic layer was washed with water and brine, dried over sodium sulfate, and solvent was removed under reduced pressure. Column chromatography was performed to isolate the product **7a** (31 mg, 29%) as yellow solid. $^1$H NMR (400 MHz, CDCl$_3$) δ: 12.27(s, 1H), 8.04 − 8.03 (m, 1H), 7.79 (s, 1H), 7.49 − 7.45 (m, 1H), 7.09 (d, J = 7.2 Hz, 1H), 6.87 (d, J = 8.8 Hz, 1H), 6.43 (t, J = 5.6 Hz, 1H), 6.35 (s, 2H), 5.66 (s, 2H), 5.29 (s, 1H), 4.94 − 4.90 (m, 1H), 4.54 − 4.44 (m, 2H), 3.79(s, 5H), 3.73-3.42 (m, 13H), 2.89 − 2.72 (m, 3H), 2.16 − 2.11 (m, 1H) ppm. $^{13}$C NMR (100 MHz, CDCl$_3$) δ: 173.5, 170.7, 170.1, 169.4, 167.7, 163.1, 152.6, 146.8, 136.1, 135.5, 134.6, 132.5, 116.7, 111.7, 110.3, 105.8, 104.7, 72.8, 70.5, 70.2, 70.0, 69.4, 56.1, 48.9, 42.5, 38.9, 34.6, 31.6, 22.8 ppm. LCMS(ESI); m/z: [M + H]$^+$ calcd. for C$_{34}$H$_{41}$N$_8$O$_{10}$, 721.29; Found 721.42. HRMS calcd. for C$_{34}$H$_{41}$N$_8$O$_{10}$ [M + H$^+$], 721.2847; found, 721.2875.

*Synthesis of 4-(4-((2,4-diaminopyrimidin-5-yl)methyl)−2,6-dimethoxyphenoxy)-N-(2-(2-((2-(2,6-dioxopiperidin-3-yl)−1,3-dioxoisoindolin-4-yl)amino)ethoxy)ethyl)butanamide (**7b**):* The procedure was analogous to that described for compound **7a**. 4-(4-((2,4-diaminopyrimidin-5-yl)methyl)−2,6-dimethoxyphenoxy)butanoic acid **4b** (51 mg, 0.14 mmol) and 4-((2-(2-aminoethoxy)ethyl)amino)−2-(2,6-dioxopiperidin-3-yl)isoindoline-1,3-dione **6a** (51 mg, 0.14 mmol) as starting materials furnished **7b** (38 mg, 38%) as yellow solid. $^1$H NMR (400 MHz, CDCl$_3$) δ: 11.07 (s, 1H), 7.53-7.49 (m, 2H), 7.33 (s, 1H), 7.10 (d, J = 7.2 Hz, 1H), 6.87 (d, J = 8.4 Hz, 1H), 6.67 − 6.55 (m, 2H), 6.31 (s, 2H), 5.77 (s, 1H), 5.02-5.00 (m, 1H), 3.99 (t, J = 5.6 Hz, 2H), 3.77 (s, 6H), 3.71-3.66 (m, 2H), 3.58- 3.55 (m, 4H), 3.43 − 3.36 (m, 3H), 2.81 − 2.73(m, 3H), 2.53 (t, J = 6.8 Hz, 2H), 2.11 − 2.04 (m, 3H) ppm. $^{13}$C NMR (100 MHz, CDCl$_3$) δ: 173.6, 173.4, 170.9, 169.5, 167.6, 163.7, 153.7, 146.8, 136.3, 135.7, 132.4, 116.9, 111.8, 110.3, 105.2, 72.3, 70.2, 68.5, 56.0, 48.6, 42.1, 39.1, 34.2, 33.2, 31.1, 26.2, 22.7 ppm. LCMS(ESI); m/z: [M + H]$^+$calcd. For C$_{34}$H$_{41}$N$_8$O$_9$,

705.30; Found 705.44. HRMS calcd. for $C_{34}H_{41}N_8O_9$ [M + H$^+$], 705.2997; found, 705.2980.

*Synthesis of 4-(4-((2,4-diaminopyrimidin-5-yl)methyl)−2,6-dimethoxyphenoxy)-N-(2-(2-(2-((2-(2,6-dioxopiperidin-3-yl)−1,3-dioxoisoindolin-4-yl)amino)ethoxy)ethoxy)ethyl)butanamide* (**7c**): The procedure was analogous to that described for compound **7a**. 4-(4-((2,4-diaminopyrimidin-5-yl)methyl)−2,6-dimethoxyphenoxy)butanoic acid **4b** (51 mg, 0.14 mmol) and 4-((2-(2-(2-aminoethoxy)ethoxy)ethyl)amino)−2-(2,6-dioxopiperidin-3-yl)isoindoline-1,3-dione **6b** (57 mg, 0.14 mmol) as starting materials furnished **7c** (46 mg, 44%) as yellow solid. $^1$H NMR (400 MHz, CDCl$_3$) δ: 12.39 (s, 1H), 7.78 (s, 1H), 7.47 (t, J = 7.6 Hz, 1H), 7.09 (d, J = 7.2 Hz, 1H), 6.88 (d, J = 8.4 Hz, 1H), 6.46 (t, J = 5.2 Hz, 2H), 6.34 (s, 2H), 5.56 (s, 1H), 5.28 (s, 1H), 4.94 − 4.90 (m, 1H), 3.95 (t, J = 4.2 Hz, 2H), 3.76 (s, 6 H), 3.71-3.59 (m, 8H), 3.52 − 3.40 (m, 6H), 2.88 − 2.74 (m, 3H), 2.45(t, J = 7.2 Hz, 2H), 2.13 − 1.98 (m, 4H) ppm. $^{13}$C NMR (100 MHz, CDCl$_3$) δ: 173.5, 173.2, 170.8, 169.4, 167.6, 163.0, 162.3, 156.3, 153.6, 146.8, 136.1, 135.6, 134.1, 132.5, 116.7, 111.7, 110.3, 105.9, 105.0, 72.0, 70.5, 70.1, 69.9, 69.3, 56.1, 48.9, 42.4, 39.2, 34.8, 33.3, 31.6, 26.2, 22.7 ppm. LCMS(ESI); m/z: [M + 2H]$^+$ calcd. for $C_{36}H_{46}N_8O_{10}$, 750.33 Found 750.46. HRMS calcd for $C_{36}H_{46}N_8O_{10}$ [M + H]$^+$, 749.3259; found, 749.3257.

*Synthesis of 4-(4-((2,4-diaminopyrimidin-5-yl)methyl)−2,6-dimethoxyphenoxy)-N-(20-((2-(2,6-dioxopiperidin-3-yl)−1,3-dioxoisoindolin-4-yl)amino)−3,6,9,12,15,18-hexaoxaicosyl)butanamide* (**7e**): The procedure was analogous to that described for compound **7a**. 4-(4-((2,4-diaminopyrimidin-5-yl)methyl)−2,6-dimethoxyphenoxy)butanoic acid **4b** (51 mg, 0.14 mmol) and 4-((20-amino-3,6,9,12,15,18-hexaoxaicosyl)amino)−2-(2,6-dioxopiperidin-3-yl)isoindoline-1,3-dione **6c** (81.2 mg, 0.14 mmol) as starting material furnished **7e** (34 mg, 26%) as yellow syrup. $^1$H NMR (400 MHz, MeOD-d4) δ:7.59 − 7.53 (m, 2H), 7.12−7.08 (m, 2H), 6.56 (s, 2H), 5.11−5.06 (m, 1H), 3.94 (t, J = 6.0 Hz, 2H), 3.82 (s, 6H), 3.74 (t, J = 5.2 Hz, 2H), 3.67 − 3.51 (m, 24H), 3.40−3.34 (m, 7H), 2.90−2.76 (m, 1H), 2.50 − 2.46 (m, 2H), 2.02−1.95 (m, 2H). $^{13}$C NMR (100 MHz, MeOD-d4) δ: 176.3, 175.1, 172.0, 171.0, 169.6, 165.0, 162.5,155.2, 154.1, 148.5, 137.5, 137.0, 136.3, 134.2, 118.6, 112.3, 108.8, 107.2, 73.6, 72.0, 71.9, 71.8,71.5, 70.9, 56.9, 43.6, 40.8, 34.7, 33.9, 32.5, 27.7, 24.0 ppm. LCMS(ESI); m/z: [M + H]$^+$ calcd. for $C_{44}H_{61}N_8O_{14}$, 925.42; Found 925.64. HRMS calcd. for $C_{44}H_{61}N_8O_{14}$ [M + H]$^+$, 925.4307; Found 925.4298.

For the *synthesis of 4-(4-((2,4-diaminopyrimidin-5-yl)methyl)−2,6-dimethoxyphenoxy)-N-(2-(2-(2-((2-(1-methyl-2,6-dioxopiperidin-3-yl)−1,3-dioxoisoindolin-4-yl)amino)ethoxy)ethoxy)ethyl)butanamide* (**7 f**), the procedure is analogous to that described for compound **7c**, with 4-(4-((2,4-diaminopyrimidin-5-yl)methyl)−2,6-dimethoxyphenoxy)butanoic acid (18 mg, 0.05 mmol) and 4-((2-(2-(2-aminoethoxy)ethoxy)ethyl)amino)−2-(1-methyl-2,6-dioxopiperidin-3-yl)isoindoline-1,3-dione (21 mg, 0.05 mmol) as starting material; furnished **7 f** (12 mg, 31%) as yellow solid. $^1$H NMR (400 MHz, CDCl$_3$) δ: 8.03(s, 1H), 7.47-7.43 (m, 1H), 7.24 (s, 1H), 7.05 (d, J = 7.2 Hz, 1H), 6.87 (d, J = 8.8 Hz, 1H), 6.81-6.78(m, 1H), 6.34 (s, 2H), 4.93-4.88(m, 1H), 3.97-3.94 (m, 2H), 3.77 (s, 6 H), 3.70-3.65 (m, 3 H), 3.59-3.52 (m, 7H), 3.44-3.42(m, 3 H), 3.15(s, 3H), 2.99 (s, 1H), 2.90(s, 1H), 2.75-2.70(m, 2H), 2.53-2.49 (m, 2H), 2.09-1.98 (m, 4H). LCMS(ESI); m/z: [M + 2H]$^+$ calcd. for C37H48N8O14, 764.35; Found 764.85. HRMS calcd for $C_{37}H_{47}N_8O_{10}$ [M + H]$^+$, 763.3410; Found 763.3415.

## Cell culture

HEK293T (ATCC) and HCT116 cells (ATCC) were cultured in complete media: DMEM supplemented with 10% fetal bovine serum (Invitrogen), 2 mM glutamine, 100 U/mL penicillin and 100 mg/mL streptomycin (all from Gibco). JURKAT (ATCC) and OVCAR8 (ATCC) cells were cultured in complete media: RPMI 1640 supplemented with 10% fetal bovine serum (Invitrogen), 2 mM glutamine, 100 U/mL penicillin and 100 mg/mL streptomycin (all from Gibco). Cells were maintained in a humidified incubator at 37 ˚C.

## Lentivirus production and generation of stable cell lines

Stable cell lines expressing eDHFR-YFP-T2A-Luciferase (eDHFR-YFP) or eDHFR-Luciferase-T2A-mCherry (eDHFR-Luc) were generated by lentiviral transduction. eDHFR-YFP-T2A-Luc and eDHFR-Luc-T2A-mCherry genes were cloned into a pTRPE lentiviral vector backbone (gift of the Milone, Riley, and June labs at Penn), and lentivirus was packaged using HEK293T/17 (ATCC) and 2nd generation packaging plasmids psPAX and pMD2 (Addgene). Target cells were transduced with lentivirus overnight in the presence of 8 µg/mL of polybrene (Millipore), washed and incubated with fresh media for 1–2 days, passaged, and were sorted on either YFP (for eDHFR-YFP) or mCherry (for eDHFR-Luc) expression using fluorescence-activated cell sorting (FACS).

## In vitro dose–response and time course assays

TMP PROTAC **7c** was solubilized in 100% DMSO to achieve a 10 mM stock solution. The stock solution of 10 mM **7c** was serially diluted in PBS (Corning) accordingly and each dose was added to the cells such that the final concentration of DMSO in cell media is <1%. Following incubation of the cells with **7c**, cells were harvested for downstream flow, luminescence, or Western blot analyses.

## Preparation of JURKAT-eDHFR-YFP for flow cytometry

$3\times10^5$ JURKAT-eDHFR-YFP cells were seeded in clear (Falcon) 12-well plates in complete media. Compound **7c** was added to each well at varying concentrations at −24, −8, and −4 h, and all samples were harvested and analyzed together on a flow cytometer (LSR II, BD) at 0 h (t = 0) to assess the degree of YFP expression.

## Preparation of OVCAR8, HEK293T, and HCT116-eDHFR-Luc for plate reader assay

$4\times10^4$ OVCAR8-eDHFR-Luc, HEK293T-eDHFR-Luc, and HCT116-eDHFR-Luc cells were plated in black wall/clear bottom (Falcon) 96-well plates in 200 µL of complete media and incubated with serially diluted compound **7c** for various durations. D-luciferin (GoldBio) was prepared in complete media and added to a final concentration of 0.15 mg/mL in each well. Luminescence was read on a plate reader (ThermoFisher Varioskan Plusplate).

## Western blotting

**Cell lysate.** Harvested cells were solubilized in radio-immunoprecipitation assay (RIPA) lysis buffer (Cell Signaling Technology) with protease inhibitor (Roche Boehringer Mannheim). The cell lysate was incubated on ice for 30 min and centrifuged at $21,000 \times g$ for 10 min (Thermo Scientific Sorvall Legend Micro 21 R). The supernatant (cell lysate) was removed and transferred to a new Eppendorf tube for later use.

**BCA assay.** Total cell protein was quantified using bicinchoninic acid (BCA) assay kit (Thermo Scientific) and bovine serum albumin (BSA) standards ranging from 10 to 0.625 mg/mL (Thermo Scientific). 3 µL of cell lysate was added into a 96-well clear plate (Falcon), mixed with 50 µL BCA reagent, and incubated at 37 °C for 30 min while shaking at 200 RPM. The absorbance of samples was measured at 480 nm on a plate reader (ThermoFisher Varioskan Plusplate). A calibration curve was developed, and samples were prepared to equal mass (mg) of total protein for gel electrophoresis.

**SDS-PAGE gel.** Cell lysate was prepared by mixing with 4 µL of loading dye and PBS to achieve equal total protein and volume across all samples. Each sample was loaded into a NuPage gel (Invitrogen) (4-12% Bis-tris) and developed in NuPage MES Running Buffer (Invitrogen/Novex). Once complete, the gel was removed and prepared for protein transfer to membrane.

**Protein transfer.** The SDS-page gel was prepared for protein transfer onto a polyvinylidene difluoride (PVDF) membrane (BioRad) activated by methanol. The sandwiched transfer cassette was loaded and developed in NuPage transfer buffer (Invitrogen/Novex) composed of 20% methanol at 4 °C for 1.5 h. The transfer was confirmed by Ponceau dye, which was washed and removed prior to incubation with antibodies.

**Antibody incubation.** PVDF membranes were blocked in 5% Milk/TBS for 1 h at RT, then rinsed gently with Tris-buffer saline (TBS) + 1% Tween (TBS-T). Next, the membrane was incubated with a primary antibody composed of 1:1000 antibody:5% Milk/TBS at 4 °C overnight. The membrane was rinsed 3x with TBS-T and 1x with TBS followed by incubation with a secondary antibody composed of 1:1000 antibody:5% Milk/TBS for 1 h at RT. Following incubation, the membrane was rinsed 3x with TBS-T and 1x with TBS and prepared for imaging.

**Western blot analysis.** Using an enhanced chemiluminescence (ECL) kit (BioRad), the PVDF membrane was treated with 1:1 mixture of the reagents and incubated for 5 min at RT. Excess liquid was removed from the membrane, which was then immobilized onto a cassette and imaged in a dark room with film.

**Western antibodies.** The following antibodies were used for Western blot analyses: COX IV Mouse mAb (Cell Signaling Technology, 11967s, Lot 3), GFP Rabbit Ab (Cell Signaling Technologies, 2555 s, Lot 6), Ikaros Rabbit mAb (Cell Signaling Technology, 14859s, Lot 1), Aiolos Rabbit mAb (Cell Signaling Technology, 15103s, Lot 4), CK1 Rabbit mAb (Cell Signaling Technology, 2655s, Lot 2), eRF3 Rabbit mAb (Cell Signaling Technology, 14980s, Lot 1), FLAG Mouse mAb (Millipore Sigma, F1804-200UG, Batch SLCK5688), Anti-mouse IgG HRP-linked Antibody (Cell Signaling Technology, 7076s, Lot 36), and Anti-rabbit IgG HRP-linked Antibody (Cell Signaling Technology, 7074s, Lot 30). Primary and secondary antibodies were applied as a 1:1000, antibody:5%milk/tbs solution.

**Detecting multiple proteins by western blot on a single membrane.** For proteins of similar molecular weight that needed to be detected from a single set of experimental cell lysates, PVDF membranes were stripped with a Urea buffer and probed again. Stripping buffer is composed of 6.5 M Urea, Tris base, pH 7.5. Prior to membrane stripping, 15 mL of Urea buffer is aliquoted and 90 μL of 2-Mercaptoethanol and microwaved for 10 s. The membrane is incubated with the buffer for 30 min then washed with TBST 3 times and TBS 1 time and subsequently blocked with 5% Milk/TBS. Following this, the antibody incubation process is repeated as described previously.

**Inhibitor test**
$5 \times 10^5$ HEK293T-eDHFR-YFP cells were seeded in a clear 6-well plate (Falcon) in complete media and incubated overnight. The next day, the cells were pre-incubated with either 500 nM epoxomicin, 25 μM hydroxychloroquine sulfate (HCS), 500 nM MLN4924, or 25 μM 3-methyladenine (3-MA) for 1 h. Following pre-incubation, the cells received either 100 nM of **7c**, 25 μM TMP, or 2.5 μM pomalidomide, and were incubated for an additional 12 h. Cells were then isolated as previously described and prepared for Western blot analysis.

**Ligand block test**
HEK293T-eDHFR-YFP cells were treated with 25 μM TMP or 25 μM pomalidomide or vehicle, directly followed by the addition of **7c**. After 24 h of incubation, using the Western blot protocol outlined above, we measured eDHFR-YFP expression with GFP Rabbit Antibody (Cell Signaling Technologies, 2555 s) and COX IV loading control with COX Mouse Monoclonal Antibody (Cell Signaling Technology, 11967 s).

**Assessing ligand mechanism with compound 7f**
HEK293T-eDHFR-YFP cells were incubated with 100 nM **7f**, 100 nM **7c** or vehicle for 24 h. Then using the Western blot protocol outlined above, we measured eDHFR-YFP expression with GFP Rabbit Antibody (Cell Signaling Technologies, 2555s) and COX IV loading control with COX Mouse Monoclonal Antibody (Cell Signaling Technology, 11967 s).

**Washout experiment**
**Kinetic readout with western blot analysis.** $3 \times 10^5$ HEK293T-eDHFR-YFP cells were seeded in a clear 12-well plate (Falcon) in complete media and incubated overnight. The next day, the cells were incubated with 100 nM **7c** for 24 h. Following incubation, media was removed by vacuum, cells were gently washed twice with PBS (Corning), and returned to culture in complete media. Cells were harvested at various time points before and after drug washout and prepared for Western blot analysis.

**Multiplexed regulation of mCherry-FKBP12$^{F36V}$ and EGFP-eDHFR**
OVCAR8-mCherry- FKBP12$^{F36V}$-EGFP-eDHFR cells were incubated with either **7c**, dTAGv-1 or both drugs for 24 h. Cell lysates were collected as described above. First, all membranes were assessed for mCherry-FKBP12$^{F36V}$ expression using anti-mCherry antibody. Once complete, the membranes were stripped as described above and probed for GFP-eDHFR expression using anti-GFP antibody.

**Kinetic readout with flow cytometry.** $3 \times 10^5$ JURKAT-eDHFR-YFP cells were seeded in a clear 12-well plate (Falcon) in complete media and incubated overnight. The next day, all wells were dosed with 100 nM of **7c**, and cells were sampled at 0, 4, 8, 12, and 24 h following incubation (1 well was sampled per time point). Following 24 h incubation, the remaining wells of cells were collected and centrifuged at 200 xg for 5 min (Thermo Scientific Sorvall Legend X1R). Cells were washed 3 times with PBS (Corning) and seeded on a new, clear 12-well plate (Falcon) in fresh complete media. The cells were sampled at 3, 6, 24, 48, and 72 h following the drug washout. All cells were fixed in 4% paraformaldehyde (PFA; Sigma) following sampling, and samples from all time points were analyzed together on a flow cytometer (LSR II, BD).

**Cell viability assay**
HEK293T-eDHFR-YFP and primary human eDHFR-FLAG T cells were plated in 96 well plates at approximately 40 E 5 cells per well in complete media. After 24 h, cells were treated with **7c** for an additional 24 or 48 h. Then using the Promega CellTiter 96® AQueous One Solution Cell Proliferation Assay, absorbance was measured by plate reader (ThermoFisher Varioskan Plusplate) at 490 nm.

**Immunofluorescence microscopy**
OVCAR8 cells expressing POI-eDHFR-FLAG constructs were cultured on round glass cover slips in complete media at about 100–200 E 5 cells per well in 12 well plates. After 24 h, media was removed, and cells were fixed using 4% formaldehyde/PBS solution for 20 min. Next cells were gently rinsed in ice-cold PBS 2× followed by blocking and permeabilizing with BSA-PBS + 0.1% Triton X (B-PBST) 30 min. This was removed and then cells were incubated Incubate with primary mouse antibody Anti-FLAG primary antibody in B-PBST for 1 h at room temperature while rocking. The primary was gently aspirated, and wells were washed with PBS for 5 min 2×. Next cells were incubated with anti-mouse AlexaFluor-488 secondary antibody in B-PBS for 1 h at room temperature in the dark. The secondary was gently aspirated, and wells were washed with PBS for 5 min 3× in the dark. Next slides were mounted on a glass slide that contained 10 μL drop of Vectashield + DAPI stain. Slides were stored overnight at 4 °C. Slides were imaged using a Zeiss LSM 880 and Zeiss LSM 980 confocal microscopes with a

×20/0.8 NA air immersion objective lens. Experiment and data collection completed as 3 biological replicates, $n = 3$.

**Immunofluorescence antibodies.** FLAG Mouse mAb (Millipore Sigma, F1804-200UG) and Goat anti-mouse pAB AlexaFluor-488 (Abcam, 150113, Batch GR3284150-1).

## Mass spectrometry proteomics

HEK293T-eDHFR-YFP and primary human eDHFR-FLAG T cells were plated in 6 well plates and were treated with 100 nM **7c** or vehicle. After 24 h, harvested cells were solubilized in 8 M urea/50 mM ammonium bicarbonate lysis buffer then sonicated using a Diagenode sonicator at medium setting for 0.5 min on and 0.5 min off for 5 min. Lysates were then treated with 1000 U of benzonase (PierceTM Universal Nuclease for Cell Lysis, 88700) for 30 min on ice. Lysates were then centrifuged at 17,000 × $g$ for 10 min and the supernatant transferred to a separate tube. Lysates were treated with 5 mM DTT for 30 min at RT followed by 10 mM IAM for 45 min at RT in the dark. Samples were then digested to peptides with sequencing grade modified typsin (Promega V5111). Peptides were cleaned for LC-MS/MS analysis using a Hamilton C18 stage tip columns and apparatus. Peptides were loaded onto the column and washed three times with 0.1% formic acid and eluted using 60% LC-MS grade acetonitrile and 0.1% formic acid. Samples were then dried via speed vacuum. Cleaned peptides were reconstituted in 0.1% formic acid and normalized to 0.33 μg/μL by A214/A280 Scopes method. Samples were loaded onto a Dionex UltimateTM 3000 LC and injected at 3 μL per sample onto a Thermo Pepmax C18 trap column and separated on a 35 cm × 75 μm I.D. laser-pulled silica column containing a 2.4 μm C18 resin packed under pressure. Separation of peptides occurred over a 2-h gradient consisting of standard proteomics mobile phase buffers (Mobile phase A: 0.1% formic acid, Mobile phase B: 0.1% formic acid and 80% acetonitrile) from 5% to 25% mobile phase B over 90 min, followed by 25% to 45% from 90 to 120 min, followed by a column wash. Peptides were ionized at 2.8 kV to a Themo Fisher QE-HFTM mass spectrometer and data was acquired using resolutions of 60k for both MS1 and MS2 and an AGC target of 1 E6 and 5 E5 for MS1 and MS2 respectively. MS2 windows were designed in a 25×24 m/z staggered window scheme for the same length as the gradient and fragmented with 28% HCD energy.

Raw files were processed using DIA-NN with standard settings and MS1 and MS2 tolerance settings at 10ppm. All other settings were kept as default. FASTA files were accessed and downloaded from Uniprot on 5/3/2020 for both the *H. sapiens* proteome and DHFR *E. coli*. Data files were imported and wrangled using RStudio. Statistical significance was determined using a two-sided Student's $t$ test with the target protein being present in all four conditions for either the HEK293T or T cell, treated and untreated samples with comparisons being limited to only within cell lines. Data were visualized via R package ggplot2. Statistical significance thresholds were considered changes greater than two-fold and a $p$-value less than 0.05.

## In vivo experiments in mice

**Intraperitoneal tumor growth and drug treatment in CD1 Nu/ Nu mice.** CD1 nu/nu female mice were injected with $10 \times 10^6$ OVCAR8-eDHFR-Luc cells intraperitoneally (IP). Tumor growth was monitored over 4 weeks (28 days) via BLI (IVIS Spectrum, Perkin Elmer) to establish baseline tumor luminescence. On days 28 and 30, mice were again imaged for baseline analysis, and mice were separated into 3 treatment groups on day 32; (1) Vehicle control (1:1 PEG and water), (2) 200 mg/kg of **7c** in 1:1 PEG and water, and (3) 40 mg/kg of TMP + 40 mg/kg of pomalidomide in 1:1 PEG and water. Mice were treated with drug or vehicle control IP and imaged by BLI 1 h following treatment. Subsequently, mice were imaged +1, 2, 3, 7, and 9 days after drug injection to monitor changes in luminescence signal.

CD1 nu/nu female mice were housed according to UPenn IACUC protocol. Time-controlled lighting on standard 12:12 light:dark cycle, 7 days a week. Ventilation was maintained at approximately 10-15 air changes per hour. Temperature was maintained between 20 and 24 °C and relative humidity was maintained between 30% and 70%.

**Compound formulation for in vivo studies.** **7c** was formulated by dissolving compound into 1:1 PEG:water. A solution of TMP and pomalidomide was generated by dissolving 1:1 TMP:pomalidomide in 49.5:49.5:1.0 PEG:water:DMSO. The vehicle contained 1:1 PEG:water.

**Bioluminescent imaging (BLI) analysis.** ROIs were drawn using Living Image Software (Perkin Elmer) around the entirety of the peritoneal cavity. Total flux in photons per section (p/s) were measured and recorded.

## Statistical analysis

All the statistical data analysis was performed on Prism 9 (GraphPad). An unpaired, two-tailed Student's $t$ test was used to assess for statistical significance between two groups, and a $p$-value of <0.05 was considered to be statistically significant.

## Experimental and technical replicates

"$n =$" refers to the number of biological replicates unless otherwise specified.

## Reporting summary

Further information on research design is available in the Nature Portfolio Reporting Summary linked to this article.

## Data availability

Data are available upon request to the corresponding author. Data generated in this study which are not included in the manuscript are provided as Supplementary Information/Source Data files. The mass spectrometry proteomics data used in this study have been deposited to the ProteomeXchange Consortium via the PRIDE[60] partner repository with the dataset identifier PXD045052. Source data are provided with this paper.

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

## Acknowledgements
The authors would like to thank members of the Small Animal Imaging Facility (Eric Blankemeyer), and the Flow Cytometry and Cell Sorting Facility. Additionally, we thank Kimberly Edwards and Gabrielle Blizard (UPenn) for helpful discussions as well as the Burslem and Farwell Labs (UPenn) for shared analytical and preparative resources.

## Author contributions
M.A.S. conceived of the project. A.R., J.D.N., J.M.E., N.S., and M.A.S. designed the molecules. All authors contributed to the experimental design. N.S. performed chemical synthesis and characterization of the PROTAC molecules. J.M.E, I.K.L, N.S., K.X., S.S., and T.N. performed in vitro experiments, analyzed data, and interpreted results. J.M.E. and R.L. performed the proteomics. I.K.L., K.X., and M.A.S. performed in vivo animal experiments. J.M.E., I.K.L., and N.S. wrote the draft of the manuscript. All authors contributed to the final paper.

## Competing interests
The University of Pennsylvania has filed pending IP (WO2022217295A1) on compounds related to this work. M.A.S., J.M.E., I.K.L., N.S., A.R., J.D.N. are inventors. M.A.S. is a co-founder of Vellum Biosciences, which is supporting the commercialization of this IP. All remaining authors declare no competing interests. Financial support includes: M.A.S. is

supported by the National Institute of Health Office of the Director Early Independence Award (DP5-OD26386), Burroughs Wellcome Fund Career Award for Medical Scientists (CAMS), and NIGMS (R01GM150804). This research was also supported by a grant from NIGMS (R35GM142505) to G.M.B. and from NCI (F31CA275040) to J.M.E.
