## [Peer Review File · Nature Communications]

REVIEWER COMMENTS

Reviewer #1 (Remarks to the Author):

Etersque and colleagues describe a new inducible protein degradation system using trimethoprim-based PROTACs and eDHFR tags. The work is original and adds to previously described systems using auxin, lenalidomide, or dTAG for induced protein degradation that are great research tools and may have clinical implications in the future. The results look solid and the authors demonstrate that the system effectively works for ectopically expressed tagged-proteins in vitro including simultaneous activity of multiple degraders and also remarkably well in vivo.

However, there are several limitations of the study:

-The effect of cpd 7c on GSPT1 at concentrations <100 nM is quite substantial and a major drawback/ limitation of a protein degrader system since it is predicted to be highly toxic. The effect on viability and proliferation as well as GSPT1 protein degradation needs to be determined in different human cell line models and primary cells.

-In order to capture the full spectrum of (wanted and unwanted) proteins degraded by 7c global assessment of protein level needs to be performed (e.g. proteomics) in treated cells. Ideally in comparison to one or several of the established systems.

-Systematic comparison with the established systems on the same protein target in different cell models is needed to demonstrate equal/ superior activity

-Given the insensitivity of mice to pomalidomide, I am wondering if this system is also working in mouse cells (cell lines/ primary)? This will have huge implications on the application of this system in genetic/ syngenic mouse models.

-All experiments were carried out on ectopically expressed tagged proteins delivered via lentiviral vectors. Important for future applications of this system is to show that this is also true for endogenous proteins and that degradation of this protein is causing a phenotype, e.g. by introducing the eDHFR-tag sequence on an endogenous protein.

-There seems to be a second manuscript by the group applying this method for CAR-T cells that is not included and the cited results from this manuscript can therefore not be evaluated.

Reviewer #2 (Remarks to the Author):

This manuscript from Sellmyer and coworkers report an efficient system for targeted degradation of proteins fused to the E. coli dihydrofolate reductase (eDHFR) by PROTACs, containing a known ligand (TMP) for this protein linked to pomalidomide.

The system is thoroughly characterised, including experiments in different cell lines and combinations with degradation of other proteins, and finally the PROTAC is shown to have efficacy in a xenograft model in mice.

This system has the future potential to be optimised for regulation of CAR expression in a CAR T-cell context.

Based on the further development potential of the work and the thorough approach, this reviewer recommends acceptance of the work after minor revision.

1) It should be noted in figure captions and experimental section how many times the WB experiments were repeated. Usually, a minimum of three individual experiments would be expected. And the full blots should be added to the Supplementary material.

2) There is a problem with the reference (Lee et al 2023) on page 14, which appears to be in a different citation format and not included in the reference list.

Reviewer #3 (Remarks to the Author):

The authors present here a new protein tag and corresponding PROTAC for targeted protein degradation, based on eDHFR. Noteworthy results include the ability to multiplex protein tags for controlled degradation of two different proteins, however, a comparison between the three tags

mentioned in the paper (eDHFR, dTAG and HaloTag) would have been beneficial. The selectivity for eDHFR tagged proteins over human DHFR is impressive and comparison to selectivity observed with these other tags would strengthen the impact of this new tag/PROTAC system. In vivo studies to show degradation of eDHFR tagged protein is also noteworthy, but there was a high level of variability in the mouse tumor size.

The use of this tag is likely to be much more impactful if it could be applied using CRISPR/Cas9 methods for endogenous proteins - despite its larger size, HaloTag has been efficiently incorporated in this way for various targets.

The authors noted degradation was not observed in HCT116 cells due to low Cereblon expression - could this tag be targeted using other E3s to address this concern? Were any additional PROTACs designed for alternative E3s i.e. VHL?

As mentioned in the manuscript, IMiD-containing PROTACs have the risk of off-target degradation of corresponding neosubstrates. The authors could consider including a phenyl glutarimide ligand to reduce these effects.

The authors demonstrate the ability to degrade various POIs with different cellular localization, yet they do not confirm that expression of the protein fusion does not have an effect on this localization. It would be beneficial to perform some microscopy studies to confirm localization, perhaps using one of the fluorescent TMP ligands. Additionally, demonstration of the use of an N- and C-terminal tag would improve the potential versatility of the tag.

Along with the competition between PROTAC 7c and pomalidomide to show required engagement of Cereblon, does competition with TMP show the same effect?

August 23, 2023
Nature Communications

Here is a point-by-point response to each reviewer's comments for a revised manuscript, entitled "**Regulation of eDHFR-tagged proteins with trimethoprim PROTACs**" to *Nature Communications*. Our revisions are described below and nearly all the requested changes have been made. Here is a summary of the key changes to the manuscript:

- We performed proteomic analysis on HEK293 and primary human T cells expressing eDHFR +/- **7c** and looked for specific and non-specific effects. We found that GSTP1 was not affected in either cell line.
- We measured the effects on cell viability *via* live/dead assay after treatment with **7c**.
- We used immunofluorescence microscopy to characterize eDHFR-POI cellular localization.

Reviewer #1

Etersque and colleagues describe a new inducible protein degradation system using trimethoprim-based PROTACs and eDHFR tags. The work is original and adds to previously described systems using auxin, lenalidomide, or dTAG for induced protein degradation that are great research tools and may have clinical implications in the future. The results look solid and the authors demonstrate that the system effectively works for ectopically expressed tagged-proteins *in vitro* including simultaneous activity of multiple degraders and also remarkably well *in vivo*.

We appreciate reviewer's insights as well as the recognition that this is "original, may have clinical implications, and it works remarkably well *in vivo*."

1. The effect of **7c** on GSPT1 at concentrations <100 nM is quite substantial and a major drawback/ limitation of a protein degrader system since it is predicted to be highly toxic. The effect on viability and proliferation as well as GSPT1 protein degradation needs to be determined in different human cell line models and primary cells.

The reviewer rightly states that GSTP1, of the glutathione S-transferase enzyme superfamily, which catalyzes the conjugation of electrophiles with glutathione in the process of detoxification, can in some contexts affect cell viability. This is especially important in the context of characterizing the true mechanism (e.g., specificity) of anti-cancer or other therapeutic effects using heterobifunctional degraders.

To assess the impact of **7c** on GSTP1, and subsequently, viability, we performed cell viability tests *via* dose-response curves after 24 and 48 h incubation with **7c** compared to vehicle. We observed no change in cell growth or viability. This data was added to **Supplemental Figures 4** and **Fig. 3** and discussed in the manuscript results and discussion section.

Although there was some GSTP1 down-regulation by western blot, we were not able to detect GSPT1 changes in HEK293T cells compared to vehicle control using proteomic experiments (**Supplemental Figure 4**). Also, we were not able to detect GSTP1 in JURKATs by Western blot, therefore, we again chose to use proteomics and primary human T cells. We found that there was no significant down-regulation of GSTP1 (**Fig. 3**). We included the new proteomic data and modified the discussion in the manuscript.

As mentioned in the response to reviewer 3, comment 3, we will investigate fluorinating the indoline benzene group which can decrease off-target binding (Burslem G. M., *ChemMedChem* **2018**), that selectively prevents GSTP1 degradation, but not the heterobifunctional molecular target. However, exploring these analogs is not the scope of this manuscript.

2. In order to capture the full spectrum of (wanted and unwanted) proteins degraded by 7c global assessment of protein level needs to be performed (e.g. proteomics) in treated cells. Ideally in comparison to one or several of the established systems.

We appreciate this comment and how it improved the manuscript. We tested for “wanted and unwanted” proteins degraded in a proteomic mass spectrometry assay in the two cell lines listed above (HEK293T and primary human T cells) after incubation with 100 nM **7c**. We found that eDHFR-YFP peptides were downregulated in HEK293T cells and eDHFR-FLAG peptides were downregulated in primary human T cells with statistical significance, however, GSTP1 was not affected. In primary human T cells, we do see downregulation in zinc fingers IKZF1 and IKZF3, as expected. This data is included in the figures and discussed in the manuscript.

3. Systematic comparison with the established systems on the same protein target in different cell models is needed to demonstrate equal/ superior activity.

The activity and use of HaloTag and dTAG systems have been well described in the literature. We chose to experimentally test TMP-PROTACs in comparison to and combination with FKBP^{F36V}-dTAG in a single cell model, to highlight their complementarity (**Figure 4**). We added a discussion of the limitations of dTAG ligand solubility

and *in vivo* ability to degrade proteins. As the reviewer is aware, dTAG-13 was used to regulate luciferase in MV4;11 liquid tumor model (Nabet, *et al*). They showed no change in luciferase signal *in vivo* after PROTAC treatment, although the controls continued to increase in luciferase signal above baseline. To state this explicitly, they did not see clear loss of luciferase signal *in vivo*, as we did (**Figure 6**). Further testing of other fusion partners (HaloTAG) is beyond the scope of this paper, however, we do discuss the merits and pitfalls of different degron systems in this manuscript in the discussion.

Nabet B, Roberts JM, Buckley DL, Paulk J, Dastjerdi S, Yang A, Leggett AL, Erb MA, Lawlor MA, Souza A, Scott TG, Vittori S, Perry JA, Qi J, Winter GE, Wong KK, Gray NS, Bradner JE. The dTAG system for immediate and target-specific protein degradation. *Nature Chemical Biology*. 2018 May;14(5):431–441.

4. Given the insensitivity of mice to pomalidomide, I am wondering if this system is also working in mouse cells (cell lines/ primary)? This will have huge implications on the application of this system in genetic/ syngenic mouse models.

As the reviewer points out, mouse cells lines including tumor lines such as multiple myeloma are insensitive to IMiD treatment as a therapeutic. In fact, humanized mice have been made, including CRBN^{I391V} which allow study of lenolidomide and pomalidomide mechanisms in mice. However, pomalidomide still binds murine CRBN, and thus heterobifunctional molecules, such as TMP-POM, will work nonetheless to degrade eDHFR fusions, despite the loss of anti-neoplastic effects of IMiDs in mice. Please see this paper for additional details: <https://www.ncbi.nlm.nih.gov/pmc/articles/PMC5912449/>:

“As expected, using dBET1, there was no Ikaros degradation in mouse T cells or mouse myeloma cell lines. However, we found that dBET1 impressively reduced the expression of BRD4 in both human and mouse cells. Therefore, proximity-associated ubiquitin-conjugating functions of mouse and human CRBN are confirmed using dBET1.”

Akuffo AA, Alontaga AY, Metcalf R, Beatty MS, Becker A, McDaniel JM, Hesterberg RS, Goodheart WE, Gunawan S, Ayaz M, Yang Y, Karim MR, Orobello ME, Daniel K, Guida W, Yoder JA, Rajadhyaksha AM, Schönbrunn E, Lawrence HR, Lawrence NJ, Epling-Burnette PK. Ligand-mediated protein degradation reveals functional conservation among sequence variants of the CUL4-type E3 ligase substrate receptor cereblon. *J Biol Chem*. 2018 Apr 20;293(16):6187–6200. PMID: PMC5912449

5. All experiments were carried out on ectopically expressed tagged proteins delivered via lentiviral vectors. Important for future applications of this system is to show that this is also true for endogenous proteins and that degradation of this protein is causing a phenotype, e.g. by introducing the eDHFR-tag sequence on an endogenous protein.

We appreciate this comment and recognize the merit of introducing the eDHFR tag with endogenous proteins. We are actively investigating using eDHFR CRISPR-knockins in several collaborations including with the Ben Black Lab (centromere organization) and the E. James Petersson Lab (proteinopathies), however this is outside the scope of this manuscript, which we believe is important to reach the scientific community as soon as possible.

6. There seems to be a second manuscript by the group applying this method for CAR-T cells that is not included and the cited results from this manuscript can therefore not be evaluated.

Reference to the CAR-T manuscript (Lee, I. K., *et al*, 2023) was removed from this publication.

Reviewer #2

The system is thoroughly characterized, including experiments in different cell lines and combinations with degradation of other proteins, and finally the PROTAC is shown to have efficacy in a xenograft model in mice. This system has the future potential to be optimized for regulation of CAR expression in a CAR T-cell context. Based on the further development potential of the work and the thorough approach, this reviewer recommends acceptance of the work.

We thank the reviewer for the supportive comments.

1. It should be noted in figure captions and experimental section how many times the WB experiments were repeated. Usually, a minimum of three individual experiments would be expected. And the full blots should be added to the Supplementary material.

We added to figure legends the number of replicates of each Western blot experiment, which were repeated at least three times; few were repeated twice. We also submitted an Excel file that includes full Western images of all blots used in this publication.

2. There is a problem with the reference (Lee et al 2023) on page 14, which appears to be in a different citation format and not included in the reference list.

Reference to the “Lee, I. K., et al, 2023” manuscript was removed from this publication.

Reviewer #3

The authors present here a new protein tag and corresponding PROTAC for targeted protein degradation, based on eDHFR. Noteworthy results include the ability to multiplex protein tags for controlled degradation of two different proteins, however, a comparison between the three tags mentioned in the paper (eDHFR, dTAG and HaloTag) would have been beneficial.

We thank the reviewer for the feedback and positive comment. We address the general idea of a comparison to dTAG in response to reviewer 1, point 3 (see above).

The selectivity for eDHFR tagged proteins over human DHFR is impressive and comparison to selectivity observed with these other tags would strengthen the impact of this new tag/PROTAC system. *In vivo* studies to show degradation of eDHFR tagged protein is also noteworthy, but there was a high level of variability in the mouse tumor size.

For clarification, these were intraperitoneal injections of human ovarian carcinoma (OVCAR8 tumor cells) that are a model of micro-metastases around the peritoneal cavity (a common mechanism of spread of ovarian cancer). Thus, the tumor size and correlative BLI signal are more variable than, for instance, a subcutaneous xenograft. However, we are comparing the mouse to itself after **7c** or vehicle administration. Therefore, the amount of luminescence signal is internally controlled and the error bars for the quantitative data are statistically significant (**Figure 6E**).

1. The use of this tag is likely to be much more impactful if it could be applied using CRISPR/Cas9 methods for endogenous proteins - despite its larger size, HaloTag has been efficiently incorporated in this way for various targets.

Thank you for the comment. We agree that knock-in of eDHFR is an important next step for our future work and we have formed active collaborations in this domain. However, this is outside the scope of this work, which shows proof-of-concept and feasibility of this important new class of regulatory PROTACs. We discuss future directions in our response to Reviewer #1 Comment 5.

2. The author's noted degradation was not observed in HCT116 cells due to low Cereblon expression - could this tag be targeted using other E3s to address this concern? Were any additional PROTACs designed for alternative E3s i.e. VHL?

We thank the reviewer for calling attention to the CRBN low HCT116 cell line. We retested **7c** in HCT116 cells and see mild downregulation (**Supplemental Figure 4**). The conclusion remains the same, that some cell lines that are CRBN deficient or low may not be the best for 7c regulation.

We also appreciate the reviewer's note consideration of alternative E3 ligase-targeting moieties. We synthesized TMP-VHL-1 using an alternative linker strategy. We then tested in HEK293T-eDHFR-YFP cells and HCT116 eDHFR-Luc cells. Protein levels in both cell lines are modestly regulated by TMP-VHL-1 similarly to CRBN. Given that this is a new linker strategy, and the focus of this paper is on the CRBN ligands, we provided the data for the reviewer below, but plan to publish this a subsequent manuscript.

[Redacted]

3. As mentioned in the manuscript, IMiD-containing PROTACs have the risk of off-target degradation of corresponding neosubstrates. The authors could consider including a phenyl glutarimide ligand to reduce these effects.

As mentioned above, to test for off-target degradation we performed mass spectrometry proteomic experiments on **7c** as compared to vehicle control in HEK293T eDHFR-YFP+ and primary human T eDHFR-FLAG+ cells, which shows the expected off-target effects at **7c** levels geared for maximal degradation of eDHFR fusion proteins (100 nM **7c**). (See data under Reviewer #1 Comment 2)

We acknowledge that there are chemical strategies to reduce the off-target effects of IMiD-containing PROTACs. For example, fluorinating the indoline benzene group can decrease off-target binding (Burslem G. M., *ChemMedChem* **2018**). Alternatively, as the reviewer suggests, phenyl glutarimide analogs of pomalidomide could also be used to target Cereblon, but reduce the propensity to target neosubstrates (Min, J., *et. al. Angew. Chem. Int. Ed.* **2021**.) We added discussion and references of these alternative molecular analogs to our future directions of study.

4. The authors demonstrate the ability to degrade various POIs with different cellular localization, yet they do not confirm that expression of the protein fusion does not have an effect on this localization. It would be beneficial to perform some microscopy studies to confirm localization, perhaps using one of the fluorescent TMP ligands.

We tested for subcellular localization of regulated proteins using immunofluorescence microscopy with anti-FLAG primary antibody and AlexaFluor-488 secondary antibody, and show proper subcellular localization of eDHFR-POI-FLAG fusions. This data is included in **Figure 4E** (discussed in lines 250-256). In **Supplemental Figure 7**, we show these data but with each fluorescence channel separated.

5. Additionally, demonstration of the use of an N- and C-terminal tag would improve the potential versatility of the tag.

We do assess both N- and C- terminally tagged proteins (see figure below). Wandless and colleagues demonstrated that eDHFR can regulate proteins on either terminus in the context of the drug-on eDHFR-DD system. Further, the Michnick lab used split eDHFR, again demonstrates the C- and N- terminal versatility of eDHFR.

Iwamoto M, Björklund T, Lundberg C, Kirik D, Wandless TJ. A general chemical method to regulate protein stability in the mammalian central nervous system. *Chemistry & biology*. 2010 Sep;17(9):981–988. PMID: 20851347

Remy I, Campbell-Valois FX, Michnick SW. Detection of protein–protein interactions using a simple survival protein-fragment complementation assay based on the enzyme dihydrofolate reductase. *Nature protocols*. 2007 Sep;2(9):2120–2125. PMID: 17853867

6. Along with the competition between PROTAC **7c** and pomalidomide to show required engagement of Cereblon, does competition with TMP show the same effect?

Thank you for this question. Using HEK cells expressing eDHFR-YFP we determined that cells treated with either TMP or pomalidomide prior to treatment with **7c** showed no degradation of the protein as determined by Western blot analysis (**Supplemental Figure 3C**).

We thank you for the opportunity to resubmit this manuscript, and look forward for your response.

Mark A. Sellmyer MD, PhD
University of Pennsylvania
Assistant Professor, Department of Radiology
Department of Biochemistry and Biophysics (secondary)
Abramson Cancer Center, Radiobiology and Imaging Program
Stellar-Chance Labs, 813A (Office), 815 (Lab)
422 Curie Boulevard, Philadelphia PA, 19104-6059
cell: 617-543-6546
office: 215-573-3212 (P), 215-573-6620 (F)
web: <https://www.med.upenn.edu/sellmyerlab/>

REVIEWERS' COMMENTS

Reviewer #1 (Remarks to the Author):

The authors responded sufficiently to most of my points of concern including performing the important proteomics experiment. There is just one but highly relevant exception that needs to be addressed referring to point 1 of my remarks:

The critical off-target of some CRBN compounds critical for toxicity is G1 To S Phase Transition 1 (GSPT1) and not GSTP1 (Glutathion S-Transferase P1). The two different proteins are confused in the reply and the revised manuscript (figure 3 and Suppl figure 5) since only GSTP1, which is less relevant, is shown in the proteomics and western blot data is shown. Instead, the effect on GSPT1 needs to be shown. The lack of toxicity in Jurkat and 293T cells (Fig 5b) is a positive finding which suggests that there is no substantial effect on GSPT1.

Besides this minor point that needs to be addressed this is now a very nice study and the described system will be of great value for the molecular biology community.

Reviewer #3 (Remarks to the Author):

The authors describe the use of eDHFR tag as a strategy to induce degradation of a POI. Notable data includes the ability to use multiple tags within one system and the in vivo data shared for their designed PROTAC.

I'm satisfied that the authors have addressed my presented questions and concerns from the original review and do not have any additional comments to add. I especially appreciate the detail they went to to describe current work on VHL-based PROTACs and the CRISPR-knock-in work - looking forward to the additional publications to come with this follow up.